# Learning the bistable cortical dynamics of the sleep-onset period

Zhenxing Hu[1]*, Manaoj Aravind[1], Xu Lei[2], J. Nathan Kutz[3‡], Jean-Julien Aucouturier[1‡]

**1** Department of Robotics and Control, Université Marie et Louis Pasteur, SUPMICROTECH, CNRS, institut FEMTO-ST, Besançon, France, **2** Faculty of Psychology, Sleep and Neuroimaging Center, Southwest University, Chongqing, China, **3** Department of Applied Mathematics and Electrical and Computer Engineering, University of Washington, Seattle, Washington, United States of America

‡ Shared senior authorship.
* zhenxing.hu@femto-st.fr

## Abstract

Humans just don't fall asleep like a log – or step-function. Rather, the sleep-onset period (SOP) exhibits dynamic and non-monotonous changes of electroencephalogram (EEG) with high, and so far poorly understood, intra- and inter-individual variability. Computational models of the sleep regulation network have suggested that the transition to sleep can be viewed as a noisy bifurcation at a saddle node which is determined by an underlying control signal or "sleep drive". However, such models do not describe how internal control signals in the SOP can produce rapid switches between stable wake and sleep states, nor how these state-space changes are translated in the macroscopic EEG. Here, we propose a minimally-parameterized stochastic dynamical model, in which one slowly-varying control parameter drives the wake-to-sleep transition while exhibiting noise-driven bistability. We provide a procedure for estimating the parameters of the model given single observations of experimental sleep EEG data, and show that it can reproduce a wide variety of SOP phenomenology. Using the model to analyze a pre-existing sleep EEG dataset, we find that the estimated model parameters correlate with subjective sleepiness reports. These results suggest that the bistable characteristics of the SOP can serve as biomarkers for tracking intra- and inter-individual variability of sleep-onset disorders.

## Author summary

Recent neuroscience research has showed a growing interest in understanding the complexity of how we fall asleep. Electroencephalographic (EEG) recordings of the sleep onset period show all the hallmarks of a noise-driven bistable system, but there currently exists no computational model that can be fitted on experimental data to understand this behavior. In this paper, we propose a minimally-parameterized model, which dynamics corresponds to the motion of

**Data availability statement:** Data availability preprocessed EEG data of 19 subjects are available at https://github.com/neuro-team-femto/cubic_sleep/tree/main/data/EEG (used for academic purposes under MIT license). Code availability for embedding extraction and parameter estimation is accessible on GitHub at https://github.com/neuro-team-femto/cubic_sleep/. The code for reproducing all the figures could be found at https://github.com/neuro-team-femto/cubic_sleep/tree/main/plots.

**Funding:** This work was supported by the MSCA Doctoral Network LULLABYTE (to JJA, ZH), the Région Bourgogne–Franche-Comté ANER action ASPECT (to JJA, MA), and the National Natural Science Foundation of China (grant 32471095 to XL). This work was conducted in the framework of the EIPHI Graduate School (ANR-17-EURE-0002 to JJA). JNK was supported in part by the U.S. National Science Foundation AI Institute for Dynamical Systems (grant 2112085) and the Air Force Office of Scientific Research (grant FA9550-24-1-0141). The funders had no role in study design, data collection and analysis, decision to publish, or preparation of the manuscript.

**Competing interests:** The authors have declared that no competing interests exist.

an noisy overdamped particle in a slowly tilting bistable landscape, as well as a way to fit it to an individual's sleep-onset EEG. We show that the fitted parameters of individual participants correlate with their subjective reports of sleepiness, suggesting that the model can capture important aspects of inter-individual variability, as well as provide potential biomarkers for sleep-onset disorders.

## Introduction

The ability of organisms to keep track of the time of day and maintain cycles of stable wake and sleep states has fascinated physiologists for a large part of the 20th century [1], and has become an iconic target of research for mathematical and dynamical-system modeling [2]. Following seminal work by Borbély, Daan and Beersma [3], mathematical models for sleep-wake regulation have traditionally included the interaction between two coupled processes: a relaxation oscillator (the homeostatic drive) by which 'sleep pressure' monotonically increases during wake and relaxes during sleep; and a circadian oscillator which modulates homeostatic thresholds approximately sinusoidally [4]. While these models are phenomenological in the sense that they aim to represent observable properties of targets without necessarily being derived from an existing underlying theory [5], they were found to be in good accordance with predictions made by more complex biophysical models of the ascending arousal system, such as Phillips' and Robinson's [6], and have been used to explain such diverse phenomena as sleep restriction experiments, the effect of caffeine, or changes in sleep patterns during development (for a review, see [7]).

Among other fascinating sleep-related dynamical phenomena, the transitional phase between wake and sleep ('sleep-onset period', or SOP) has attracted recent attention in the neuroscience community [8,9]. While long regarded as a monotonic process akin to " flicking a switch" or, alternatively, a long sequence of successive substates [10], the SOP is now studied as a continuous, dynamic process that fluctuates progressively, but non-monotonically, between wake and sleep, and little is known about its heterogeneity both within and between individuals [9,11]. At the surface electroencephalogram (EEG) level, the wake-to-sleep transition is primarily marked by the progressive disappearance of the EEG alpha rhythm, and exhibits all the hallmarks of bistable behaviour (Fig 1-top). However, while it is increasingly suspected that the internal dynamics of the SOP has both cognitive and clinical significance in subsequent sleep and wake periods [12,13], there exists little mechanistic understanding of what produces such patterns and their heterogeneity. Even at a basic phenomenological level, the SOP and its associated first stage of sleep (N1) remains the period with the lowest inter-scorer agreement and classification accuracy in both humans [14] and machines [15].

Our work builds on a line of biophysical models that frame the SOP as a stochastic bistable system. For instance, Yang et al. [16] demonstrated how the Philipps–Robinson (PR) model of all-day sleep regulation [6] can, near the wake-to-sleep transition, be reduced to a one-dimensional normal form equivalent to the motion of a

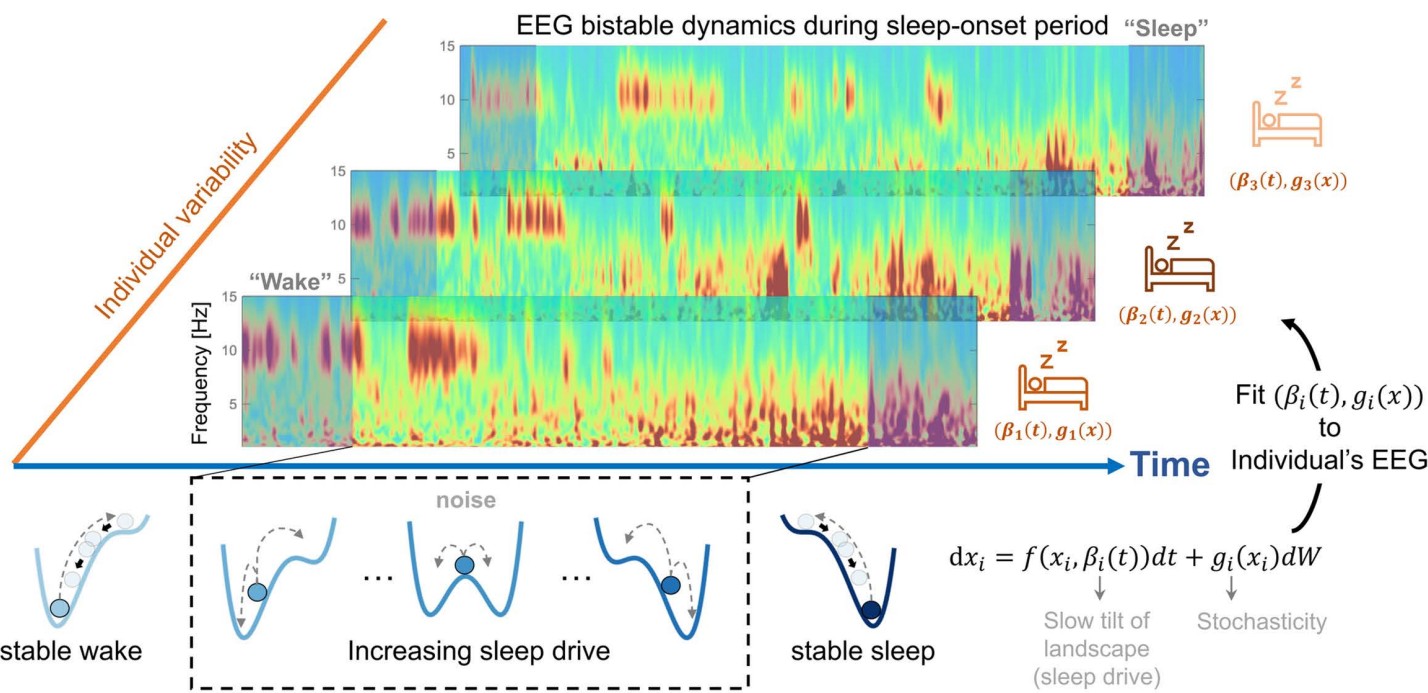

**Fig 1. Study overview.** The sleep-onset period (SOP) has a strongly bistable phenomenology, marked by a non-monotonous decrease of the EEG frequency and high inter-individual variability, seen here in three illustrative wavelet spectrograms (**top**). We model the bistable cortical dynamics of the SOP by introducing a minimally-parameterized stochastic dynamical system, which dynamics corresponds to the motion of an noisy overdamped particle in a slowly tilting bistable landscape (**bottom**). We provide a procedure to estimate model parameters given individual observations of experimental sleep EEG data (**right**), which allows us to test whether model parameters correlate with clinical feature. Sleep icon adapted from the SVG Repo, "Sleeping In Bed," CC0 License; resized and recolored by the authors.

particle in a cubic potential. This reduction yields analytical predictions for fluctuation statistics (e.g., spectral width) near the fold transition and underscores the role of stochasticity in phenomena such as flickering and variable transition timing. Subsequent work, including Fulcher et al. [17], further showed how increased noise levels (resulting from the modeling of reduced activity in orexinergic neurons) affect transition timing and switching behavior. However, while these models highlight the importance of stochasticity and time-varying drive, they lack a systematic method for inferring parameters or noise magnitudes from experimental data.

By contrast, Li et al. [9] recently proposed a deterministic bifurcation-based framework that fits empirical EEG data. Their approach involves dimensionality reduction of ~50 EEG features to construct a one-dimensional embedding representing the "sleep distance" between the instantaneous pre-sleep EEG state and sleep. They then fit a deterministic, non-autonomous bifurcation model to this trajectory. However, because their focus is on predicting the tipping point, they explicitly filter out stochastic fluctuations from both the embedding and the model. Our contribution bridges these two approaches: we develop a stochastic bistable model of SOP that accommodates time-varying drive and noise, while also providing a rigorous inference procedure to extract subject-specific parameters from EEG recordings. This enables us to quantify inter-individual variability, addressing a key direction noted by Yang et al. [16], and to potentially link these parameters to clinical conditions such as insomnia or narcolepsy, all while retaining the functional role of fast fluctuations in SOP dynamics.

To formalize this viewpoint, we propose a minimally parameterized stochastic dynamical model in which the system's state is governed by a bistable potential landscape (Fig 1). In previous work, a similar form was used to model how

external drivers control switches between the two neural states that correspond to forward and backward motion in C. Elegans [18]. Here, the system's state is assumed to move on a continuum between two potential basins correspond to the 'wake' and 'sleep' states and, instead of using an external control signal to actuate the state into sleep, we let a putative 'sleep drive' tilt the landscape towards the sleep basin (Fig 1, left to right), with the result of making noise-driven transitions to sleep not inevitable, but increasingly more likely, over the SOP. This model takes a multiscale view of SOP in which a slowly varying drive reshapes the landscape stability while fast fluctuations generate the observed intermittent excursions and reversals. This perspective treats the system's stochasticity as functionally informative rather than as observation noise that should be filtered out.

To fit our model to individual EEG data, we construct a generalized, low-dimensional representation of the SOP EEG by performing a low-rank SVD decomposition of the EEG spectrogram and use the linear interpolation between the two wake and sleep SVD modes to represent the dynamics linked to the transition. Related low-dimensional spectral decompositions have been used in other experimental settings. For example, Miller et al. [19] learn principal spectral components (PSC) via PCA from an ensemble of PSD snapshots and project the wavelet-based spectrum to the $1^{st}$ PSC to obtain real-time spectral fluctuations to capture the temporal movement trajectories of different individual fingers. While the phenomenology differ (event-locked motor behavior versus sleep-onset period), these prior results support the general principle that EEG dynamics can often be captured by a small number of spectral patterns. Then, we provide and validate a procedure for fitting our dynamical system model to the trajectory of that interpolation, and show that its dynamics can reproduce a wide-variety of SOP phenomenology. Finally, using the model to analyze a preexisting sleep (nap) EEG dataset, we test whether the estimated model parameters correlate with subjective sleepiness reports collected around the nap.

## Results

### The bistable characteristics of the SOP are preserved in a low-rank embedding of the EEG spectrogram

Sleep-onset periods (SOP) in our EEG dataset of healthy adults have a typical, strongly bistable phenomenology, marked by a non-monotonous decrease of the EEG alpha (8–12Hz) frequency [8]. In a typical participant (Fig 2a-top), the alpha component exhibits a prolonged, relatively stable amplitude early in the analysis window (here, from about 0s to 100s). This alpha activity gradually becomes more transient and intermittently suppressed, indicative of a "bistable" pattern in which the signal switches between high and low amplitude on a short timescale. In the transitional period (inside the dashed line), there are rapid switches between alpha component and the low-frequency component. Beyond 300s, the alpha band is largely diminished and replaced by an increasingly dominant low-frequency (0.5–4 Hz) component, reflecting the transition from wake into sleep-like regimes.

We construct a generalized, low-dimensional representation of the SOP spectrogram by computing the singular value decomposition (SVD) of the spectrogram separately in the initial wake and final sleep segment (see *Materials and Methods*). Both SVDs are typically rank-1 (see **Fig A in S1 Appendix**), with dominant modes $U_w^1$ and $U_s^1$ respectively (Fig 2a-middle). We then generate an embedding $\mu(t)$ of the SOP spectrogram by projecting the spectrogram on $U_w^1 - U_s^1$. Coefficient $\mu(t)$ maps how strongly the EEG at each moment aligns with the wake versus sleep mode. This coefficient neatly compresses the broad changes seen in the high-dimensional time-frequency plot - including shifts in dominant frequency modulation - into a one-dimensional time series (Fig 2a-middle).

This low-dimensional representation provides a state-space for our model. Specifically, we fit below our general dynamical systems model to the trajectories of the $\mu(t)$ embedding. Conversely, the embedding also allows us to project state-space dynamics back into observation (EEG) space, using the linear interpolation $\mu U_w^1 + (1 - \mu)U_s^1$ (Fig 2a-bottom). Although some finer details are inevitably lost in this low-dimensional representation, the major transitions and overall shifts from wakefulness to sleep are captured by this low-rank approximation. It also captures important inter-individual variations (Fig 2-b), which motivates the need to fit model parameters at the individual level and look for potential associations of these parameters with individual sleep characteristics.

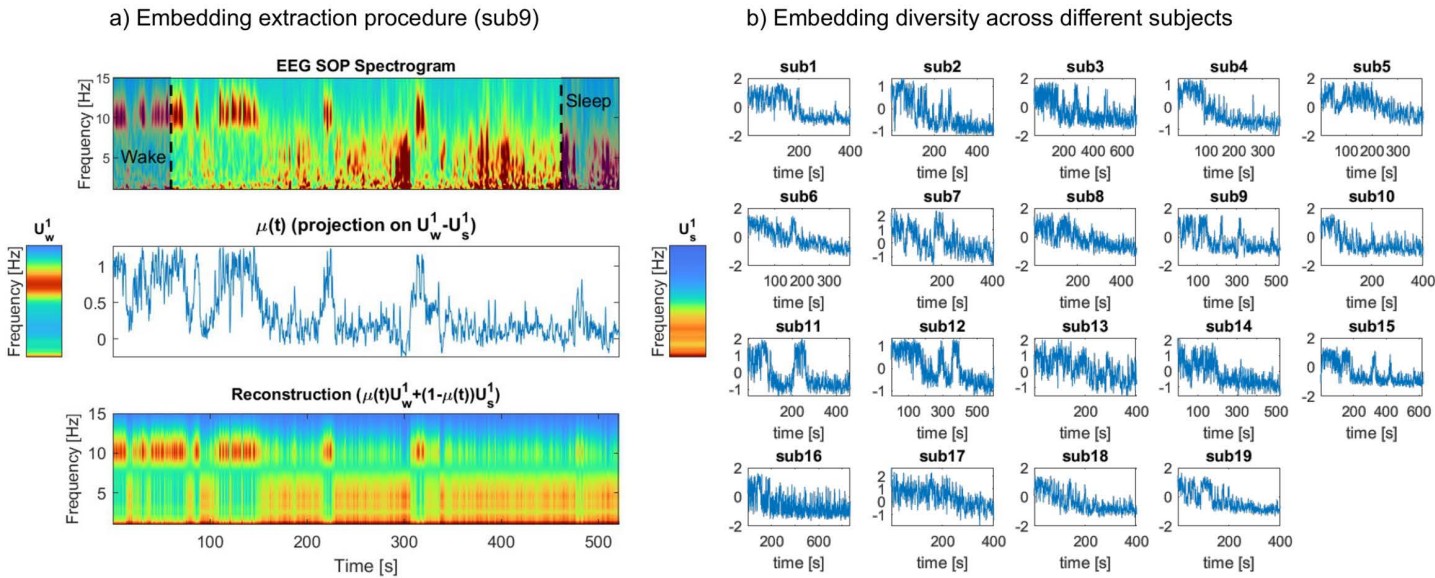

**Fig 2. Bistable dynamics are manifested in a low-rank embedding of the SOP EEG spectrogram. (a) single-subject example**. **Top:** Spectrogram representation of a single-channel (Oz, occipital, median) EEG recording of the SOP, from wake (left) to sleep (right), for one illustrative participant. Grayed windows identify the initial stable wake and final stable sleep phases, and dashed lines represent the start and the end of transitional period. **Middle:** Dominant SVD modes for the wake ($U_w^1$, **left**) and sleep states ($U_s^1$, **right**) extracted from the initial and final windows. A low-dimensional representation $\mu(t)$ is obtained by projecting the normalized spectrogram to the principal direction ($U_w^1 - U_s^1$), and captures the major transitions and shifts from wakefulness to sleep. **Bottom:** Reconstructed spectrogram obtained from the $\mu(t)$ embedding by linear interpolation between the wake and sleep mode. **(b)** $\mu(t)$ embeddings extracted from N = 19 healthy participants in our test dataset (see *Materials and Methods*).

## Changes to two model parameters reproduce a wide variety of SOP phenomenology

To model the dynamics of $\mu(t)$, we use a Langevin model [20] where the system evolves according to both deterministic forces (the potential landscape) and random noise. The system's behavior is shaped by a cubic, double-well potential landscape representing two stable states separated by a saddle, plus additive noise $n(t)$ (see Eq. 1 and *Materials and Methods*). The potential has two stable fixed points at $x = \pm 1$ and a single unstable fixed point at a time-varying location $\beta(t) \in [-1, 1]$, which is parameterized by a slowly-drifting hyperbolic tangent function with slope $\alpha$ and offset timing $t_0$ (Fig 3). The additive noise $n(t)$ is modeled as a Gaussian distribution with mean zero and standard deviation $\sigma$. This setup allows the model to capture both the deterministic tendency to settle into stable states and stochastic transitions driven by noise, and is a canonical model for many systems in, e.g., optics or atomic physics, where the nonlinear dynamics between two stable states drives transitions between states [21,22].

By controlling how quickly the landscape changes ($\alpha$) and how strongly the dynamics is subjected to stochasticity ($\sigma$), the model can capture a range of transition behaviors, from monotonous gradual shifts to abrupt, noise-driven back-and-forth switches. Fig 4 illustrates the influence of a change of parameter on simulations of the model given the same initial value and the same random seed of the per-timepoint noise distribution. In general terms, larger $\alpha$ values tend to induce earlier transitions to sleep (compare Fig 4-top and middle rows), and larger $\sigma$ values lead to increased flickering between the wake and sleep states before settling into one (compare Fig 4-middle and right columns). However, the influence of the two parameters is not entirely independent, nor linear. First, as $\alpha$ grows, the model exhibits a saturation effect in which, after a certain point, transition to sleep does not occur earlier for larger values of $\alpha$ (Fig 4-left column, middle and bottom rows). Second, at fixed levels of $\alpha$ (e.g., Fig 4-top row), increasing $\sigma$ level can induce faster transitions. Finally, at large $\alpha$ values, the pronounced "tilt" in the landscape can keep the system in the sleep state after

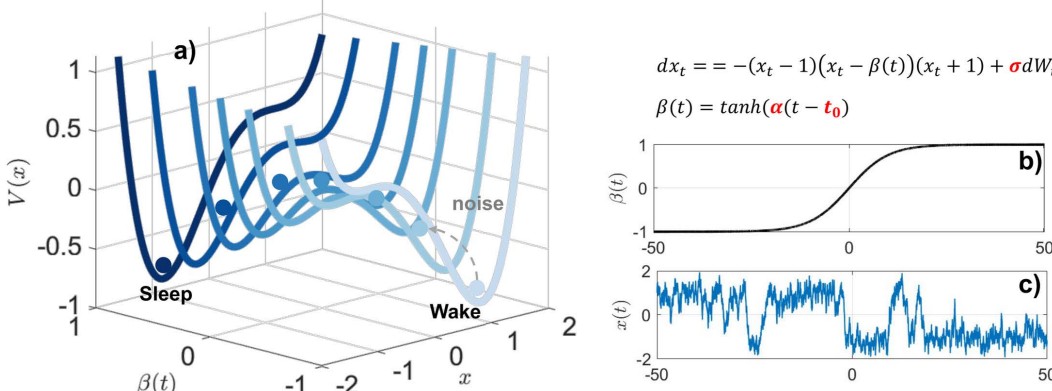

**Fig 3. Schematics of the time-varying stochastic bistable model. a)** The potential functions, $V(x)$, transitions from a "wake" basin (right) to a "sleep" basin (left) as $\beta$ shifts from -1 to +1. **b)** $\beta(t) = tanh(\alpha(t - t_0))$ governs the gradual tilt of the landscape over time. **c)** A representative trajectory $x(t)$ simulated from the stochastic differential equation Eq. 1).

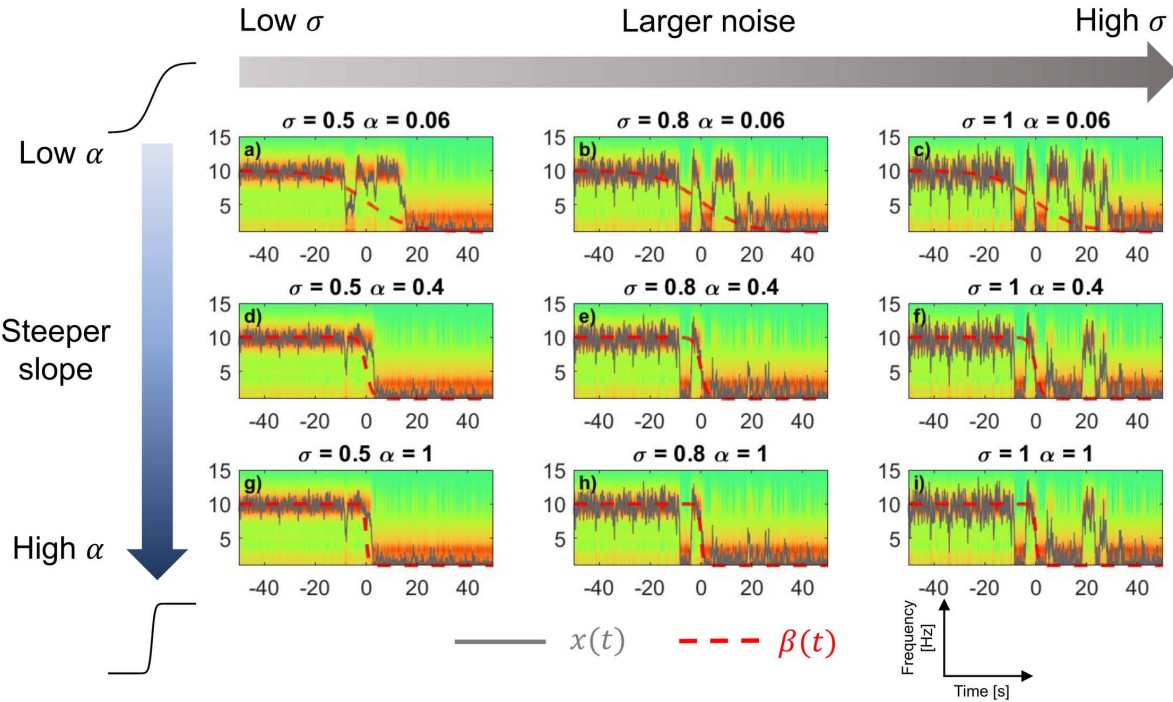

**Fig 4. Interactions between the rate of landscape change ($\alpha$) and noise ($\sigma$) in wake-to-sleep transitions.** Each panel shows a simulated spectrogram over time, with $\alpha$ increasing from top to bottom (0.06, 0.4, 1) and $\sigma$ increasing from left to right (0.5, 0.8, 1). In general terms, larger $\alpha$ values tend to induce earlier transitions to sleep (compare top and middle rows), and larger $\sigma$ values lead to increased flickering (compare middle and right columns). All $x(t)$ (grey) are normally stochastic, but simulated here with the same random seed for the purpose of comparison. $\beta(t)$ (red, dashed) is normalized in amplitude, also for visualization. EEG embedding similar as Fig 2.

transitioning, even when higher noise levels are present. The wide variety of this phenomenology in state-space can be explained by the interplay between the landscape and stochasticity, which potentially lead to the empirical inter-individual variations seen in Fig 2-b.

To further illustrate how $\alpha$ and $\sigma$ jointly influence the number of bistable jumps in simulations of the model, we quantified the number switches ($N_{sw}$) in simulated trajectories as the number of crossings of the state $x(t)$ across the zero midpoint (half-way between the two stable points at $x = \pm 1$). Fig 5 shows the switch statistics (mean and standard deviation of $N_{sw}$) from 100 simulated trajectories across a grid of $(\alpha, \sigma)$ values. The mean number of switches (Fig 5-left) increases with $\sigma$ (left to right), but this effect is strongly modulated by $\alpha$. When $\alpha$ is small (slow landscape tilt), the system remains near the bistable/low-threshold barrier for longer, allowing stochastic forcing to kick the state back and forth repeatedly, thus producing more switches. When $\alpha$ is larger (faster tilt), the transition window is shorter and the trajectory switches more quickly, limiting the number of back-and-forth transitions even when there is a larger $\sigma$. The nonzero standard deviation (Fig 5-right) across repetitions reflects trial-to-trial variability due to stochastic properties of the system.

### Model parameters can be recovered from a single experimental trajectory of the system

Infering likely values for parameters $\alpha$, $t_0$ and $\sigma$ given a single realization, i.e., one participant's SOP EEG spectrogram is made difficult because the model is (1) stochastic (so a single value of $\sigma$ can lead to an infinite number of realizations) and (2) non-stationary (so a given intermediate value $\beta_i = \beta(t_i)$ can only be fitted given one time sample of the observed trajectory). In Section *Materials and Methods*, we provide a Markov Chain Monte Carlo (MCMC) formulation of the parameter-fitting problem. Here, we use simulated data to evaluate how the procedure recovers the parameters under both stationary and non-stationary assumptions, and then illustrate parameter estimation with one representative example of experimental SOP data.

**Stationary landscape:** when the double-well potential is stationary (i.e., $\beta(t) = \beta$), the system's trajectory only reflects noise-driven fluctuations within a fixed landscape. Under these assumptions, we simulated 30 random realizations ($x(t)$ trajectories) for every pair of values $\beta \in \{-0.8, -0.6, -0.4, -0.2, 0, 0.2, 0.4, 0.6, 0.8\}$ and

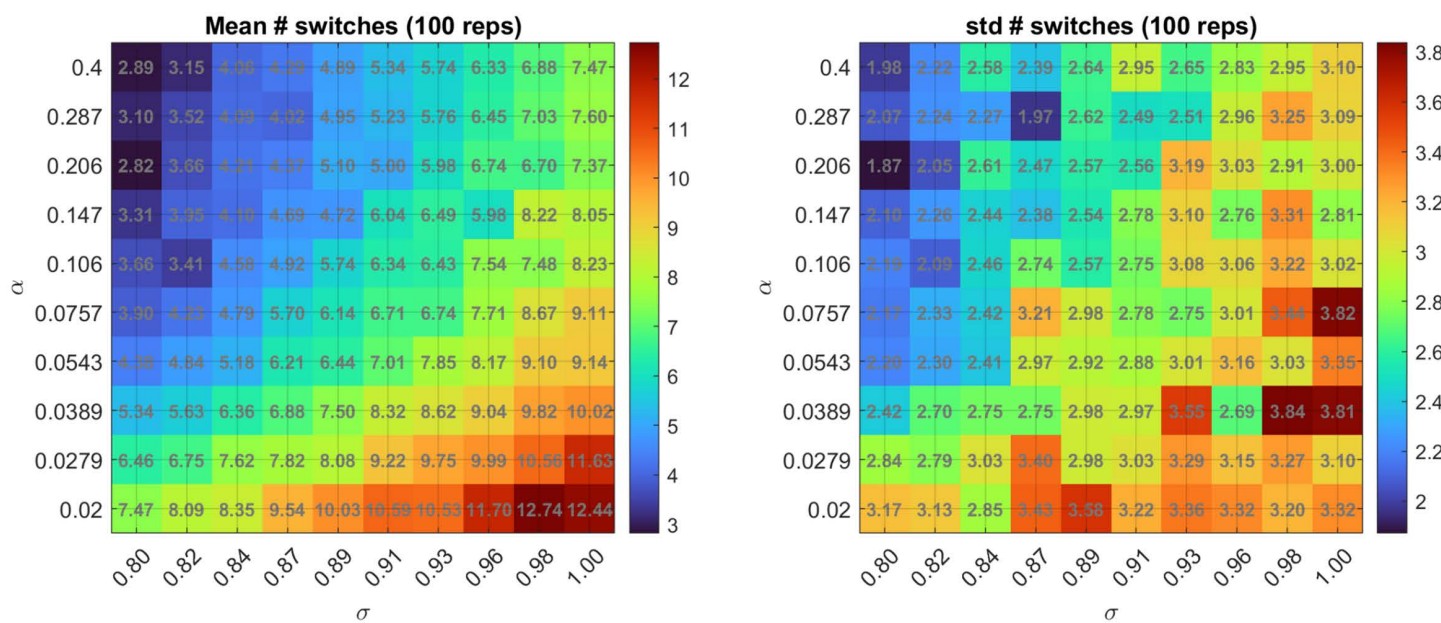

**Fig 5. Switching statistics across a grid of $(\alpha, \sigma)$ values.** Mean (left) and standard deviation (right) of the number of switches computed from 100 simulated trajectories per parameter pair. For each trajectory, we simulate the cubic SDE with the specified $(\alpha, \sigma)$. Switches are detected using a robust zero-crossing rule applied to the simulate state $x(t)$. The heatmaps illustrate an interaction between $\alpha$ and $\sigma$: increasing $\sigma$ tends to increase switching while increasing $\alpha$ shortens the bistable window and reduces opportunities for repeated back-and-forth switches. The non-zero standard deviation reflects trial-to-trial variability inherent to stochastic dynamics.

$\sigma \in \{0.2, 0.3, 0.4, 0.5, 0.6, 0.7, 0.8, 0.9, 1.0\}$ and evaluated recovered parameters for each trajectory with MCM. The procedure was able to recover both $\beta$ and $\sigma$. For $\beta$, posterior-mean estimates over the 30 trajectories showed good agreement with the true $\beta$ values (Fig 6-left), but the standard deviation over individual trajectories, although relatively moderate, was non-negligible, suggesting that some degree of degeneracy (i.e., different $\beta$ values may yield similar trajectories). For $\sigma$, MCMC estimates closely tracked true values (Fig 6-right), showing both low bias and low variance. This likely reflects that the underlying likelihood function is more sensitive to changes in $\sigma$, making noise-related parameters easier to pin down compared to the possibly overlapping trajectories that can occur for varying $\beta$.

**Non-stationary landscape:** In scenarios with a time-varying landscape, the procedure needs to estimate not only noise-level $\sigma$ but also $t_0$ (the tipping point at which the potential starts to change) and $\alpha$ (the steepness of that change). As before, we simulated 30 random trajectories for every value of $\alpha \in [0.05, 1.5]$ (logarithmically-spaced), $t_0 \in \{40, 50, 60\}$ and and $\sigma \in \{0.2, 0.3, 0.4, 0.5, 0.6, 0.7, 0.8, 0.9, 1.0\}$, and evaluated recovered parameters with MCMC estimation.

As in the fixed-landscape scenario, parameter $\sigma$ was accurately recovered for each individual simulation (Fig 7-b) with minimal standard deviation across multiple realizations of the same noise level (Fig 7-b). Also, $t_0$ estimation (Fig 7-a) was largely accurate, only deviate slightly upward when the true $t_0 = 40$. This bias is possibly due to the normal prior centered at $t_0 = 50$, as well as less sensitivity of likelihood to changes of $t_0$. Despite this bias, the ordering of the estimated $t_0$ still reflects the true ordering (i.e., the rank relationship is preserved), which is sufficient for distinguishing earlier from later transitions. Similarly for $\alpha$ (Fig 7-c), the MCMC procedure overestimated small and underestimated large $\alpha$ values, although it preserved ordering (Spearman r = 0.996 across whole range of $\alpha$ values). Notably, as $\alpha$ increases beyond approximately $1 - 1.5$, the estimated values begin to plateau, indicating a saturation effect. This is

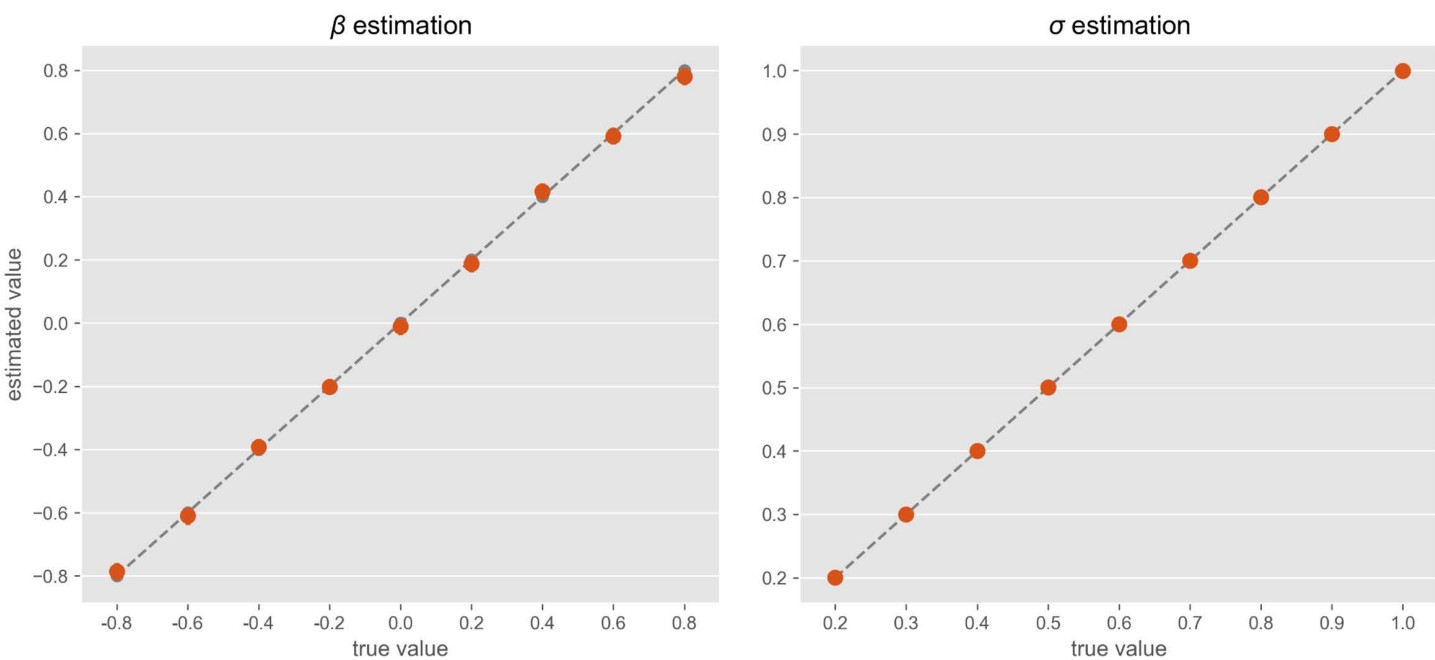

**Fig 6. Model parameter estimation for simulated single trajectories under the stationary landscape assumption ($\beta(t) = \beta$). Left:** 30 trajectories were simulated for every value of $\beta \in \{-0.8, -0.6, -0.4, -0.2, 0, 0.2, 0.4, 0.6, 0.8\}$. Distributions of MCMC estimates for $\beta$ (y-axis) are plotted against the true values (x-axis). Error bars indicate the 95% confidence interval of the mean estimates across multiple trajectories of the same $\beta$ value. **Right:** 30 trajectories were simulated for every $\sigma \in \{0.2, 0.3, 0.4, 0.5, 0.6, 0.7, 0.8, 0.9, 1.0\}$. Distributions of MCMC estimates for $\sigma$ (y-axis) are plotted against the true values (x-axis). Error bars are barely visible across multiple trajectories of the same $\sigma$.

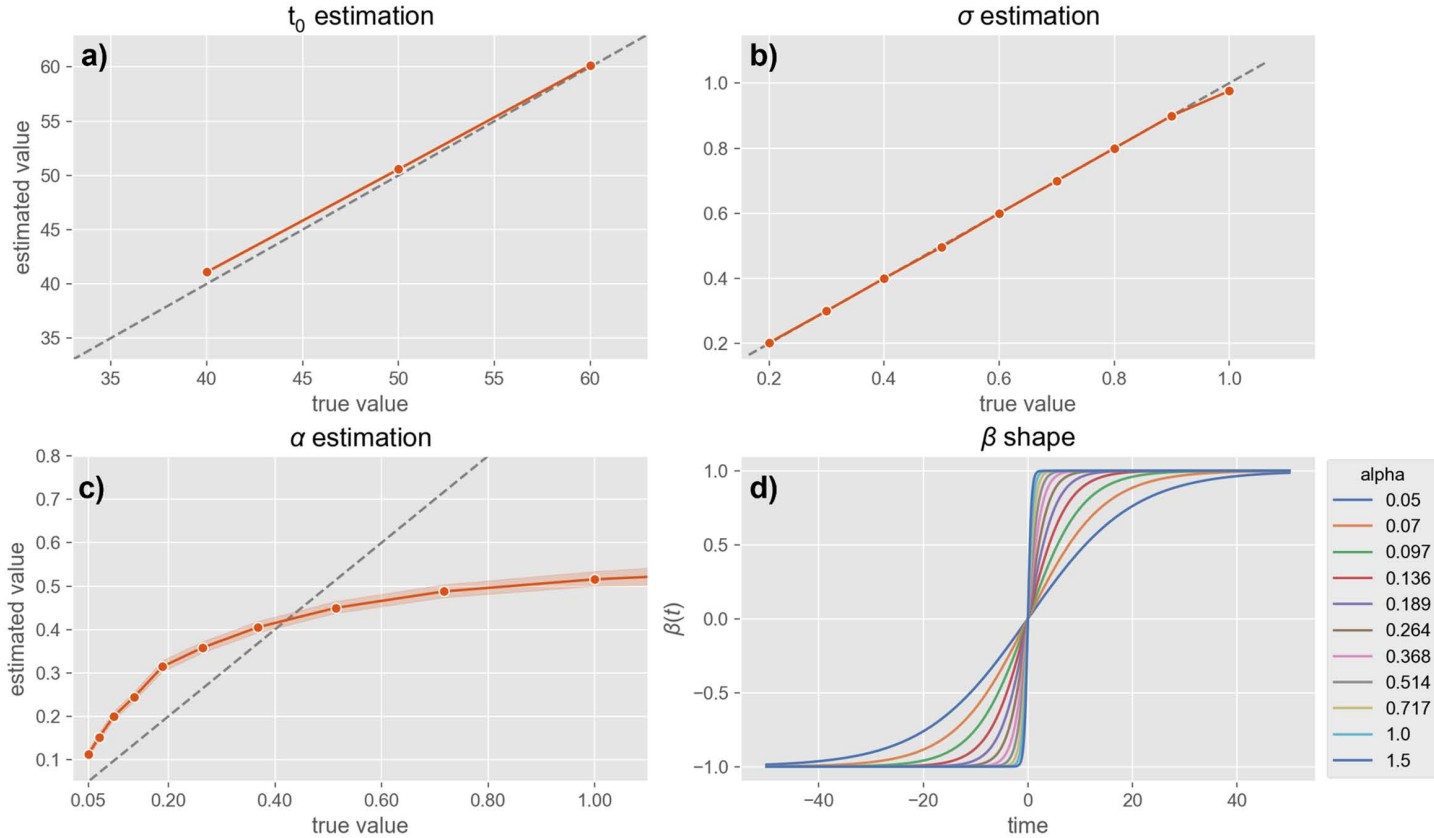

**Fig 7. Model parameter estimation for simulated single trajectories under the varying-landscape, non-stationary assumption. a):** 30 trajectories were simulated for every value of $t_0 \in \{40, 50, 60\}$ and recovered MCMC estimates for $t_0$ (y-axis) are plotted against true values (x-axis). Error bars (barely visible) indicate confidence interval across multiple realizations of the same $t_0$, and dashed line indicates perfect estimation. **b).** 30 trajectories were simulated for every value of $\sigma \in \{0.2, 0.3, 0.4, 0.5, 0.6, 0.7, 0.8, 0.9, 1.0\}$, and MCMC estimates (y-axis) are plotted against true values (x-axis). Error bars (barely visible) indicate the 95% confidence interval of the mean estimates across the multiple realizations of the same $\sigma$, and dashed line indicates perfect estimation. **c):** 30 trajectories were simulated for every value of $\alpha \in [0.05, 1.5]$ (logarithmically-scaled), and MCMC estimates (y-axis) are plotted against true values (x-axis). Error bars indicate the 95% confidence interval of the mean estimates across multiple realizations of the same $\alpha$, and dashed curve indicates perfect estimation. **d):** Varying $\beta(t)$ shape (x-axis:time in units of model integration instead of seconds) as a function of slope $\alpha$, illustrating a saturation effect at larger $\alpha$ values.

possibly because, as $\alpha$ becomes large, the slope in the underlying *tanh* function saturates (Fig 7-d) due to the inverse relationship between the time-scale of $\beta(t)$ and $\alpha$, plausibly making it more difficult for the estimation procedure to distinguish further increases in $\alpha$.

**Validation with experimental data**: Finally, we illustrate here parameter estimation, and the associated reconstructed EEG spectrogram on one illustrative example of experimental SOP data, recorded from one participant (female, 19; see *Materials and Methods*).

To compare the fitted model with the observed data, we used posterior predictive check by generating 4000 trajectories from the joint posterior distribution over parameter sets, and computed two metrics for goodness of fit: Kullback-Leiber (KL) divergence, which measures how closely the statistical distribution of a single simulated trajectory resembles that of the real EEG embedding, and Root Mean Square Error (RMSE), which reflects point-wise temporal alignment between model output and data. Neither metric alone is perfect; KL divergence ignores temporal correlations in time, whereas RMSE does not capture model stochasticity due to its point-wise nature.

We display the distribution of both metric scores (Fig 8-a). The distribution of KL-Divergence and RMSE over 4000 simulated trajectories of the estimated models shows that most KL values cluster near 0.16, and RMSE around 0.87, indicating that for many model realizations from the posterior point clouds, the distribution of states is moderately close to the empirical data. The sampled trajectory with minimal KL-distance to the true trajectory over the set of 4000 trajectories is obtained for KL = 0.06. Visually, this trajectory has a similar distribution of $x(t)$ to the original embedding (Fig 8-b), confirming that the minimal-KL solution indeed approximates the overall dwell-time distribution of the embedding $\mu(t)$. The trajectory corresponding to minimal RMSE $x_{min\_RMSE}(t)$ over the sampled set of trajectories aligns relatively well with the original $\mu(t)$ (Fig 8-c), and the real embedding remains with the 10%-90% range of trajectories simulated from the joint posterior distribution. The corresponding reconstructed spectrogram in Fig 8-g can be compared with the original SVD embedding in Fig 8-f, confirming the model's ability to generate realistic spectrograms.

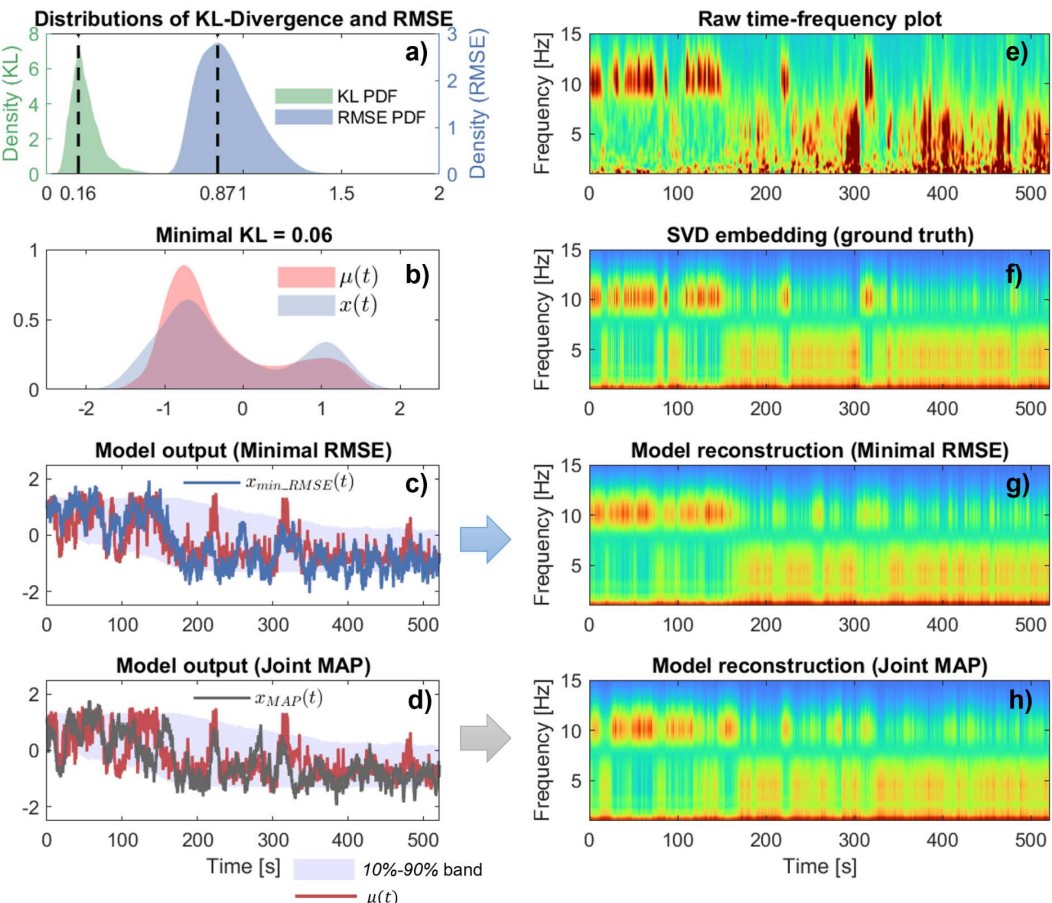

**Fig 8. Illustration of fitted parameters and model reconstruction on experimental EEG data. (a)** Distributions of the Kullback–Leibler (KL) divergence (computed on distributions pooled across all time points) and Root Mean Sqaure Error (RMSE) between the original trajectory and 4000 simulated trajectories from the posterior distributions of estimated model. Solid lines identify the simulated trajectories with the mode KL, and mode RMSE, over the random set. **(b)** Comparison of the probability density function of the minimum-KL trajectory (blue) with that of the original trajectory $\mu(t)$ (red). **(c)** Time-domain comparison of the minimal-RMSE solution (blue) with the original embedding $\mu(t)$ (red). Shaded areas correspond to the 10% - 90% amplitude span of solutions from the 4000 sampled trajectories. **(d)** Time-domain comparison of the Joint MAP solution (dark gray) with the original embedding $\mu(t)$(red). Shaded areas correspond to the 10% - 90% amplitude span of solutions from the 4000 sampled trajectories. **(e)** Wavelet spectrogram of the original participant EEG. **(f)** Reconstructed spectrogram from the original $\mu(t)$ embedding, obtained using the linear interpolation $\mu U_w^1 + (1 - \mu)U_s^1$ (similar to Fig 2a)-bottom). **(g)** Reconstructed spectrogram from the minimum-RMSE solution of panel **c. (h)** Reconstructed spectrogram from Joint MAP solution of panel **d.**

Similarly, the simulated trajectory with joint maximal posterior density (joint MAP) to the experimental data $x_{min\_RMSE}(t)$ is illustrated in Fig 8-d, and the corresponding spectrogram reconstruction in Fig 8-h. The Joint MAP reconstruction replicates global spectral transitions and captures the intermittent switching phenomenon. While the trajectory does not match point-by-point, due to the stochastic nature of the model, it captures the bistable dynamics and eventual transition to sleep-like states.

Besides KL-divergence and RMSE, we also compute the distribution of per-timepoint likelihood as well as as noise observation likelihood (see *Materials and Methods*) for each of the 19 participants in our test dataset (Fig 9; see also **Text A in S1 Appendix** for an analysis of the detailed timecourse of likelihood within participant). The ratio of the per-timepoint observation-noise and cubic transition likelihood is informative of how the MCMC procedure attributes events in the experimental data to either $\sigma$-driven transitions in the cubic dynamics (i.e., amplitude increments on $\dot{x}_t$), or to $\sigma_{obs}$-driven observation noise (i.e., amplitude increments on $x_t$ that operate outside of the state dynamics. The mode value of the distribution of transition likelihood is indicative of the model's goodness of fit for individual participants, i.e., how well the participant's embedding correspond to our assumption of stochastic bistable dynamics. Two participants (Sub17: M = 0.62 and sub13: M = 0.59) appear to have relatively low likelihood (<10%-percentile), and the embeddings of these two participants indeed do not entirely fit our expected phenomenology (slow, monotonous drift with no evident bistable jumps, refer to Fig 2).

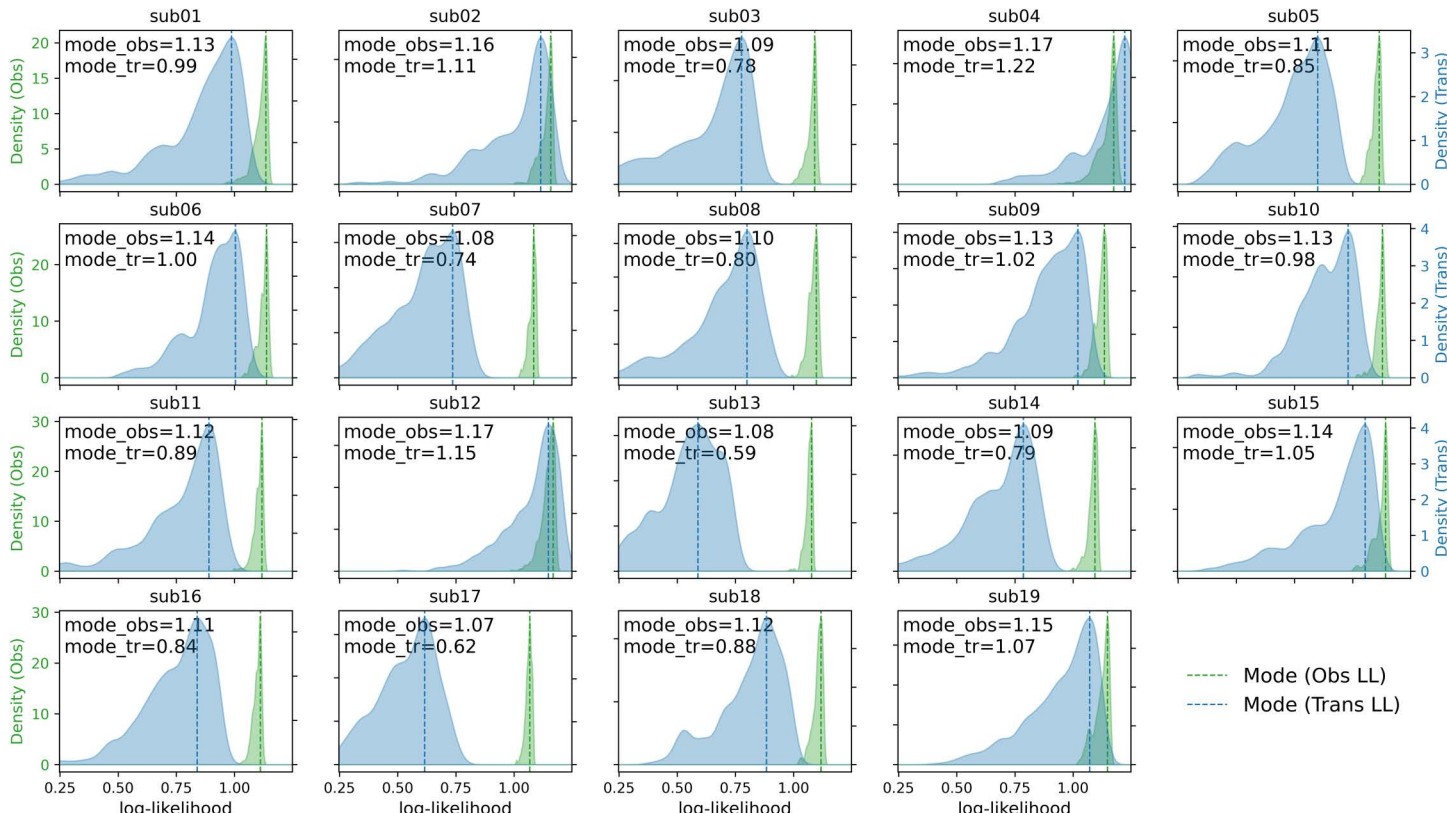

**Fig 9. KDE of per-time observation-noise and transition log-likelihood.** For each subject, we firstly compute the per-time point observation-noise and transition log-likelihood for all 4000 posterior draws, and then take the mean-value across draws at each time point, yielding a single time series of observation and transition log-likelihood values. We then aggregate these values across time and plot their kernel density estimate (KDE). The dashed vertical line marks the mode of the KDEs.

Finally, to quantify whether our model estimation on real-data is able to preserve the bistable jumping phenomenology of the data, we used same procedure as Fig 5 to quantify and compare switching statistics between the modeled embedding and simulated trajectories of the correspondind model. For each subject in our test dataset, we simulated 100 trajectories using that subject's fitted parameters, and compared the average number of switches over these simulations with the actual number of switches in the subject's SOP (Fig 10). There was a positive statistical association between real and simulated switching (Pearson $r = 0.51, p = 0.036$; with two poor-fitted subjects Sub17 and SUb13 excluded), which supports the idea that the fitted model correctly captures cross-subject variability in switching (when the subject's data is compatible with cubic dynamics in the first place).

### Estimated model parameters correlate with subjective ratings of sleepiness

The sleep (nap) dataset used in this study contains subjective reports of sleepiness by N = 19 participants on two different measures: the Stanford Sleepiness Scale (SSS) and Karolinska Sleepiness Scale (KSS).

In an exploratory manner, we investigated whether estimated parameters from the model (slope and tipping point of sleep drive $\alpha$ and $t_0$, noise $\sigma$) correlate with any of these characteristics. We found that the pre-nap sleepiness level on the Stanford Sleepiness Scale (SSS) shows a significant positive rank relationship with $\alpha$ (Spearman r = 0.65 for $\hat{\alpha}$ (Fig 11-a) and $\hat{\sigma}$ (Fig 11-b); both p-value < 0.05/3 after Bonferroni correction. Correlations with pre-nap KSS scores were consistent, albeit not significant: $\hat{\alpha}$ r = 0.33, p = .16; $\hat{\sigma}$: r = 0.43, p = .07. There was no statistical correlation between model parameters and subsequent KSS/SSS ratings after the nap. Note that all effects were tested here with rank (Spearman) correlations, in order to be robust to possible non-linear scaling in the estimation of $\alpha$ (Fig 7).

We conducted a number of confirmatory analyses. First, we verified that these correlations were not driven by the two participants (17,13) who had poor model fit as revealed by the model's transition likelihood (Fig 9). It wasn't the case, as both correlations remained statistical after we excluded these 2 participants: $\hat{\alpha}$ r = 0.61, p = 0.009; $\hat{\sigma}$: r = 0.60, p = 0.011 (see

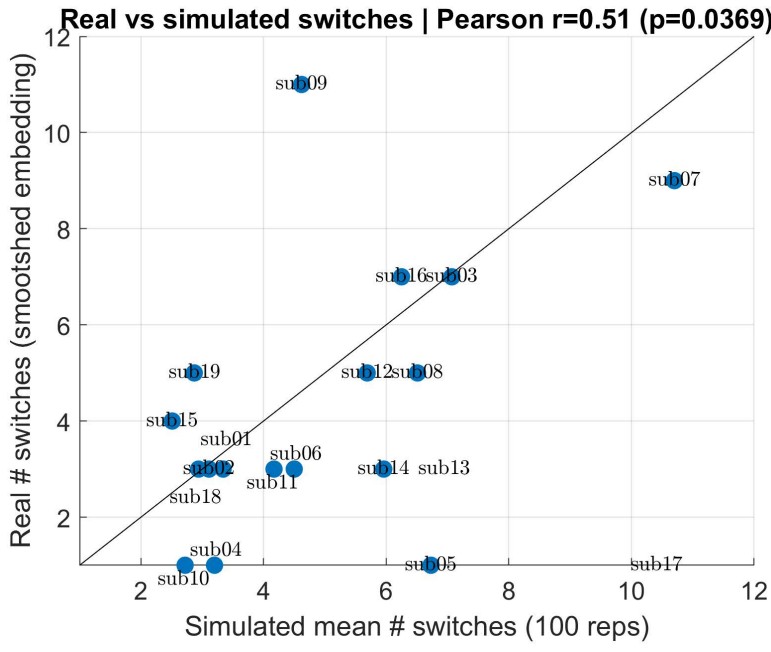

**Fig 10. Switch counts from real data and simulations using fitted ($\alpha, t_0, \sigma$).** Real switch counts versus simulated switch counts obtained by averaging over 100 simulated trajectories per subject using that subject's fitted parameters. Solid line: y = x line.

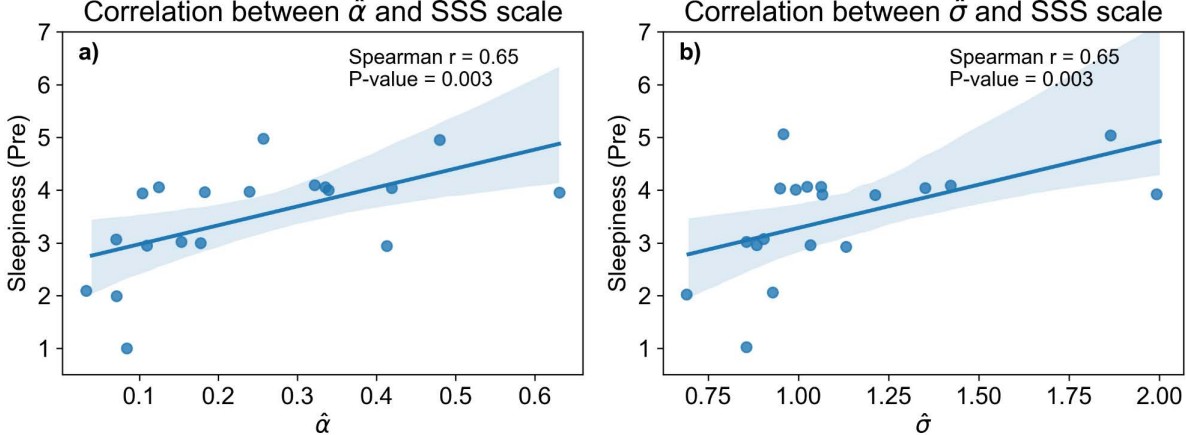

**Fig 11. Correlation of fitted model parameters with subjective sleepiness report.** (a-b) Scatter plots and linear fit model of parameters (left: $\hat{\alpha}$, right: $\hat{\sigma}$) and subjective assessment of pre-nap sleepiness of Stanford Sleepiness Scale (SSS).

**Fig B in S1 Appendix**). Second, we validated that the correlations were robust across different $\sigma_{obs}$ levels - a MCMC parameter governing how much the latent state is alowed to differ from the observed data (see *Materials and Methods* and **Fig C in S1 Appendix**). Finally, although both correlations were significant, collinearity between $\alpha$ and $\sigma$ ($\sigma \sim \alpha$: $\beta$ = 1.135, $t(17)$=2.74, $p$ = .014) makes it difficult to conclude that both parameters are truly independently predictive of subjective sleepiness. First, while both parameters are statistically predictive of sleepiness when tested individually with ordinary least-square (OLS) regressions ($SS \sim \alpha$: $\beta$ = 3.59, $t(17)$=2.91, $p$ = .01; $SS \sim \sigma$: $\beta$ = 1.64, $t(17)$=2.65, $p$ = .017), neither of them are in a joint OLS model ($SS \sim \alpha + \sigma$; $\alpha$: $\beta$ = 2.48, $t(16)$=1.72, $p$ = .11; $\sigma$: $\beta$ = 0.971, $t(16)$=1.38, $p$ = .18), which translates the model's incapacity of conclusively attributing variance to one or the other predictor. Second, both model comparisons ($SS \sim \alpha$: $AIC$ = 49.9 *vs* $SS \sim \sigma$: $AIC$ = 51.0) and residual tests ($SS \sim residual(\alpha \sim \sigma)$: $\beta$ = 2.48, $t(17)$=1.45, $p$ = .16 *vs* $SS \sim residual(\sigma \sim \alpha)$: $\beta$ = 0.97, $t(17)$=1.14, $p$ = .27) indicate weak evidence that $\alpha$ may be more predictive that $\sigma$, but we do not think that evidence is conclusive.

The positive correlation between alpha and subjective sleepiness suggests that higher sleep propensity may lead to a more pronounced or rapid transition, which appears intuitive (although a larger number of participants would be obviously needed to confirm the interpretation). The positive correlation between estimated noise-level and subjective sleepiness before the nap suggests that participants who felt sleepier also exhibited higher intrinsic noise levels. In particular, while that increase of $\sigma$ may in general lead to more numerous switches (Fig 5), including more backward sleep-to-wake transitions - which seems counter-intuitive, the simultaneous increase of $\alpha$ in the context of high sleepiness may contribute to counter-balance the number of switches. More generally, this potential role of noise, if confirmed, opens in our view an alternative mechanistic view of the wake-to-sleep transition beyond the conventional rising-sleep-drive model, linking internal neural noise to behavior aspects.

## Discussion

### Context and contribution

The sleep-onset period (SOP) is increasingly recognized as a dynamic transition rather than a simple binary switch [8,9]. Yet traditional analyses, including in the clinic, often reduce it to a single point, typically the first appearance of stage N1 [23]. Even more refined staging methods, such as the Hori classification [24], do not fully encompass the fluctuating continuum during SOP. In response, our study introduces a new modeling framework with three main contributions: (1)

a data-driven embedding strategy for high-dimensional EEG time-frequency signals, paired with a parsimonious bistable stochastic model governed by a single, slowly varying parameter that drives the wake-to-sleep transition; (2) a parameter-estimation procedure that is validated in simulated trajectory and real data given single trajectory, thereby allowing systematic quantification of intra- and inter-individual differences in SOP dynamics; and (3) exploratory evidence that estimated model parameters may correlate with subjective ratings of sleepiness around the nap, suggesting a potential utility for differentiating subtypes of sleep-onset difficulties.

Regarding the first contribution, we introduce a parsimonious embedding strategy based on linear interpolating the first SVD mode of wake and sleep states. This embedding not only reduces dimensionality but also preserves key features of the SOP trajectory. If needed, this embedding procedure can be extended to feature sets beyond spectrogram amplitude, including, e.g., phase or connectivity measures [9]. Then we model the SOP index ($\mu(t)$) with a minimal, parameterized stochastic model in which a slowly changing potential landscape and a tunable noise term together produce a wide range of SOP phenomena, including smooth drift, early or delay switch, as well as noise-driven flickering. Unlike prior models [16], which assume near-equilibrium conditions with negligible noise-level, our approach explicitly incorporates both time-varying parameters ($\beta(t)$) and stochastic fluctuations, allowing us to characterize the interactions between landscape and stochastic forcings during SOP. In future work, this modeling strategy could be extended to other transitions between states (e.g., other sleep stages, or the opposite transition from sleep-to-wake) provided they also exhibit intermittent switches. Inspired by recent works about closed-loop auditory stimulation [25,26], one could also potentially incorporate external control signals (e.g., sound) input within the derived state-space, with the application, e.g., to facilitate the process of falling asleep.

Fitting parameters of stochastic model given a single observed trajectory is challenging. Here, we employ a Markov chain Monte Carlo (MCMC) approach to fit its parameters, the slowly varying "sleep drive" and the noise term. We firstly validate the fitting in simulated settings, then apply it on real EEG recordings. The simulated-data experiments reveal that, while certain parameters (e.g., $\alpha$, $t_0$) can exhibit moderate variability, their rank ordering across individuals is largely preserved, thus allowing reliable comparisons of inter-individual differences.

Our exploratory correlation analysis suggests that, once inferred from data, model parameters can potentially serve as biomarkers for tracking intra- and inter-individual variability in sleep-onset disorders. Testing this hypothesis rigorously will require a large cohort of patients and comparing individuals with conditions like insomnia, delayed sleep phase, or narcolepsy to healthy sleepers. For example, our model may predict that patients with insomnia may have abnormally low or high noise levels, or a slower drift of the sleep drive ($\beta$), whereas narcolepsy or sleep-deprived individuals might show an abnormally steep drive toward sleep. Such applications could yield quantitative indices for diagnosing and tracking these conditions, complementing existing clinical scales.

## Comparison with previous work

Our findings extend and integrate several threads of prior research on sleep onset dynamics and modeling. Early mechanistic frameworks of sleep-wake regulation (e.g., the two-process model and "flip-flop" switching circuits) established the concept of a bistable control of sleep and wake states, but these models usually involve many variables and parameters with the lack of linking to macroscopic measurements, making them difficult to fit directly to EEG data. On the other end of the spectrum, data-driven approaches have been developed to track sleep onset. For example, Prerau et al. (2014) [11] used a statistical dynamic model to compute a continuous probability of wakefulness by combining EEG and behavioral measures, improving the temporal precision of SOP tracking over traditional sleep-stage scoring, but failed to provide mechanical insight. However, contrary from these approaches, our model explicitly captures the SOP dynamics through a physical description based on stochastic dynamical systems. By gradually adjusting the position of the central barrier between wake and sleep attractors and systematically varying the noise level, our model effectively captures the continuous and stochastic nature of sleep-onset phenomena observed empirically, including intermittent reversals or "flickering" between wake-like and sleep-like states.

A recent work by Li et al. [9] also studied the sleep-onset periord (SOP) using dynamical system perspective, and we view our work as complementary to theirs. Conceptually, both approaches leverage low-dimensional structure in EEG-derived features from one single channel to track progression through the SOP. However, Li et al. primarily targeted the final no-return tipping point near the end of SOP and applied a heavily temporal smoothing on their derived sleep-distance (1-min moving median on a 6-s feature updated every 3s). This operation yielded an almost monotonic embedding suited for identifying a robust group-level tipping signature. To facilitate a direct comparison, we reproduce their sleep-distance measure by applying their released analysis code to our recording (see **Fig D in S1 Appendix**). While both measures roughly capture relatively similar fast dynamics, temporal smoothing in the work of Li et al. removes almost all intermittent jumps.

In summary, our modeling goal differs. We aim to explain the full SOP progression at finer temporal resolution, where the data frequently exhibit intermittent, bistable switches. Preserving these fast transitions motivates a stochastic formulation [17] in which intrinsic fluctuations interact with a slowly varying sleep drive, rather than a deterministic trajectory fitted to a temporally smoothed signal. This difference in target phenomenon and time-scale has downstream implications for variability: the framework of Li et al naturally emphasizes stability of the tipping point within individuals, while plausibly compressing between-subject differences. In contrast, our framework explicitly represents inter-individual differences through a subject-specific dynamical parameters (e.g., different temporal profiles of sleep-drive ($\beta(t)$) and intrinsic noise-levels ($\sigma$)), and treats variability as a meaningful feature of SOP dynamics rather than observation noise that should be filtered out. The fact that, in our work, the estimated strength of noise $\sigma$ correlates with subjective parameters like sleepiness suggests that internal noise may indeed have functional importance in the process of falling to sleep. While we are not trying to speculate where that source of stochasticity comes from (but see Fulcher et al. [17] on narcolepsy linking frequent bistable transitions between states with reduced activity of orexinergic neurons of the lateral hypothalamus), we at least suggest it should be a target for future research.

### Limitations and future directions

This work has a number of limitations, which we describe here and for which we suggest potential strategies to address them in future work.

1) **Parameter estimation in stochastic system**: Estimating parameters in stochastic models is challenging. Firstly, a large stochastic perturbation can obscure the underlying system dynamics, making it difficult to accurately estimate the deterministic term. Second, fitting the parameters of a stochastic model to a single trajectory (one SOP per subject) is inherently difficult [27] because the variability observed in one time series may be attributable to multiple distinct parameter sets. Finally, this challenge is amplified in non-stationary systems with time-varying parameters [28]. In future work, we will leverage the neural posterior estimators to match the time-varying, noisy, and single-realization SDE case in our sleep-onset model, which is grounded in recent methodology development of simulation-based inference [29,30].

2) **Embedding strategy and dimensionality**: Our current 1-D embedding leverages the evidence of rank-1 structure in wake and sleep states (see **Fig A in S1 Appendix**) as well as the idea of mutual inhibition: interpolating between wake and sleep modes ($\mu U_w + (1-\mu)U_s$) captures how the dominant subspace in EEG spectral content shifts from wake to sleep states. To clarify what $\mu(t)$ intends to model, one can compare the spectrogram reconstructed from the $\mu(t)$ embedding (obtained by multiplying the per-timepoint of $\mu$ with $U_w^1 - U_s^1$), with a reconstruction using a truncated SVD of the whole spectrogram (Fig 12). Contrary to the individual wake and sleep states, the spectrogram SVD is typically rank-2 (Fig 12-top, right), with a first frequency mode ($U_1$) capturing the broadband power-law-like spectral structure (see ‘ **Fig E in S1 Appendix** for all participants), whose temporal weights $V_1^T$ captures total spectral power (Fig 12-middle, right) (see **Fig F in S1 Appendix** for all participants), and a second component $V_2^T$ empirically equivalent

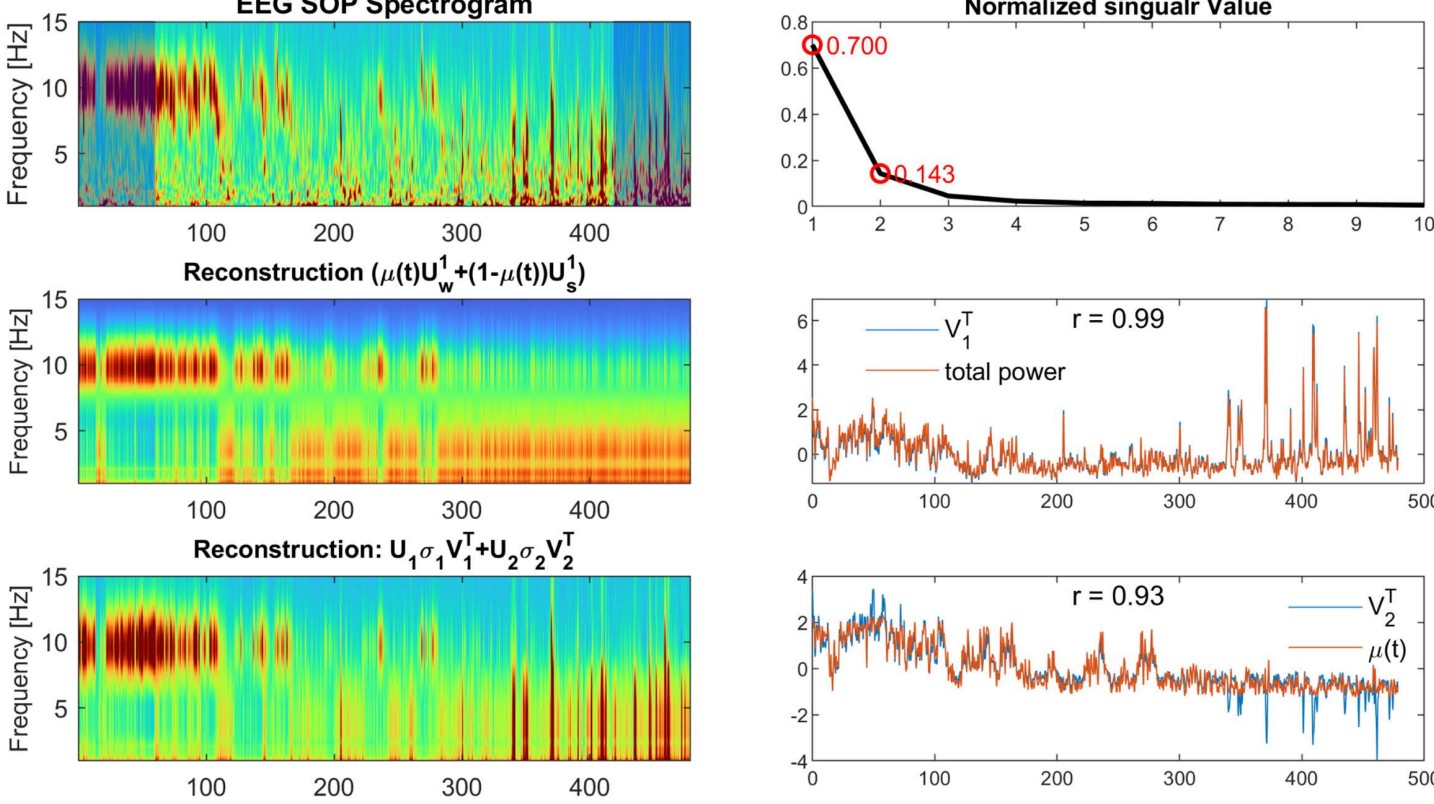

**Fig 12. Relating between 1-D transition coordinate $\mu(t)$ to a Rank-2 SVD decomposition of the full SOP spectrogram (single-subject example).** *Left column:* Top, the wavelet-based representation of the SOP spectrogram. Middle, reconstruction using 1-D embedding ($\tilde{S}(f,t) = \mu(t)U_w^1(f) + (1 - \mu(t))U_s^1(f)$). Bottom, a rank-2 SVD reconstruction of the full SOP spectrogram ($\tilde{S}(f,t) = U_1\sigma_1 V_1^T + U_2\sigma_2 V_2^T$). *Right column:* Top, explained variances of the normalized singular values from the SVD on the full SOP spectrogram. Middle, the first right-singular vectors $V_1^T$ closely tracks total spectral power (sum across frequencies) ($r = 0.99$). Bottom, the second right-singular vectors $V_2^T$ closely tracks the transition coordinate $\mu(t)$ ($r = 0.93$).

to $\mu(t)$ (Fig 12-bottom, right; see **Text B in S1 Appendix** for an analytical explanation). One can see that adding the leading component $V_1^T$ improves spectrogram reconstruction (Fig 12-bottom,left), but it mainly fills in the details of power variability within-state (e.g., large slow-wave bursts in sleep). In other words, $\mu(t)$ primarily reflects changes in spectral shape across regimes and, by design, does not explicitly represent fast within-regime amplitude modulation. Consequently, broadband power fluctuations (captured by the leading global SVD component / total power), including transient bursts within wake or sleep, are treated as residual variability rather than as an additional modeled state.

It is not known a priori whether total spectral power and $\mu(t)$ are in fact dynamically coupled, nor the form of such coupling (e.g., state-dependent, non-autonomous, or mediated by unobserved drives). Estimating a coupled 2-D model directly from noisy, nonstationary time series is therefore ill-posed without stronger structural priors. For these reasons, we only focus here on a minimally parameterized and interpretable 1-D transition coordinate, while recognizing that a 2-D extension that incorporates power dynamics may be an important direction for future work.

3) **Assumption of one global sleep-drive:** Our current model makes a modeling assumption that one global sleep-drive dictates the wake-to-sleep transition, and does not attempt to dissociate the influence of homeostasis and circadian

rhythm. Our current formulation of the sleep drive, $\beta(t)$, is a single, monotonic, exogenous parameter. While this simplification captures the idea that the overall drive increases during the sleep-onset period, it does not differentiate between distinct physiological components that are known to co-modulate sleep transitions. In neural systems, control signals are often shaped by both a deterministic, feed-forward drive and a state-dependent feedback mechanism. Although the classical circadian rhythm is typically a long-term process, the embedding developed here suggests that during sleep onset, a fast, deterministic component may emerge that prepares the neural system for transition. Without providing a physiologically grounded explanation of what $\beta(t)$ represents, our current formulation may oversimplify this complex interplay. In future work, one could aim to disentangle these components, potentially by leveraging dynamic system identification methods (e.g., dynamic SINDy [31]) that can infer separate contributions from feed-forward and state-dependent drives. This refinement will potentially enhance their biological interpretability.

## Materials and methods

### Ethics statement

The study was approved by the Ethics Review Committee of the Faculty of Psychology, Southwest University (approval no. H18058). All participants provided **written** informed consent prior to participation and had complete knowledge of the present study protocol. All procedures followed the Declaration of Helsinki.

### Dataset

N = 37 healthy college students from Southwest University in Chongqing (male: 17; female: 20) were enrolled in the study. Inclusion criteria included: no addiction to tobacco or alcohol, no substance abuse, no coffee or other functional drinks in the week before the experiment; no history of neurological or psychiatric diseases; participants needed to have a nap habit (>4 /week and each nap lasting for >30 min); a regular work and rest schedule maintained for 1 week before the experiment (the time to fall asleep no later than 00:00 hours [midnight], the time to wake up between 06:30–08:00 hours, and a total sleep duration of ( $6.5 - 8$ hr); no day–night reversal behaviour; and no crossing time zone behaviour. The subjects completed the Stanford Sleepiness Scale (SSS) and Karolinska Sleepiness Scale (KSS) before nap. Then the subjects had their EEG electronics connected and then took a nap. About 90 min later, the experimenter awoke the subjects and instructed them to complete the scale package (SSS and KSS) one more time, and then again 30 minutes after waking up. Due to a partial lack of sleep scales report, data from N = 19 subjects were finally included.

### EEG recordings

The experiment used a 63 Ag/AgCI electrode cap (Brain Products GmbH, Gilching, Germany) based on the extended 10–20 international electrode position system. Two additional electrodes were used as reference and ground, and the online reference was Fcz The electro-oculogram (EOG) was recorded using two electrodes, one electrode below the left eye and the other outside the tail of the right eye. The EEG signal was recorded with a sampling rate of 500 Hz. Before the experiment, it was ensured that impedance were $< 5k\Omega$ for all electrodes.

In all the paper, a single EEG electrode was used for all subsequent analyses, the central-occipital electrode (Oz). Oz was selected as it reliably captures the prominent alpha-band (8–12 Hz) activity, then a 0.5 - 30 Hz band-pass filter (4-order Butterworth zero-phase filter) was applied, serving as anti-alias filter and removing high-frequency artifact. After that, the signal was down-sampled down-to 100 Hz. To transform the signal to time-frequency space, we used the analytic (complex) Morlet wavelet (cmor1-1.5: band-width parameter = 1; center-frequency = 1.5 Hz), with defined frequency range of interest to span 0.5 Hz up to 20 Hz, and subdivided that interval into 200 equally-spaced frequency bins. This wavelet-based time-frequency representation prepared for the embedding extraction procedure.

## SOP window definition

From each individual participant's EEG wavelet spectrum, we identified the time window in which the participant transitions from wakefulness to sleep by examining the ratio of delta-band (0.5–4 Hz) $amp_\delta$ to alpha-band (8–12 Hz) amplitudes $amp_\alpha$. The start of transition ($t_{start}$) was defined as the earliest continuous block of time where $\frac{amp_\delta}{amp_\alpha} > 1$ for more than one minute. The end of transition ($t_{end}$) was identified as the earliest continuous block of time where $\frac{amp_\delta}{amp_\alpha} > 1$ for more than two minutes. We then defined the SOP window as [$t_{start}$−200s., $t_{end}$+200s.]. Within this window, we labelled the first 1 minute as "wake" and the last 1 minute as "sleep". Although this definition is entirely heuristic, we used it consistently across all individuals. All individual embeddings can be found in the (Fig 2-right). Individual SOP windows and spectrograms for all subjects can be found in **Fig G in S1 Appendix**.

## Embedding extraction

Using the SOP window, we then construct a generalized, low-dimensional representation $\mu(t)$ of the SOP spectrogram by computing the singular value decomposition (SVD) of the time-frequency (TF) representation separately in the "wake" and "sleep" segments on the Oz EEG channel to obtain the first principal modes, $U_w^1$ and $U_s^1$. We then generate the embedding $\mu(t)$ by projecting the spectrogram on $U_w^1 - U_s^1$. To do so, we first normalized each spectrogram bin in the SOP window to the unit norm, then projected onto ($U_w^1 - U_s^1$) to obtain a scalar $\mu(t)$. To scale the $\mu(t)$ to the same range of model output, a linear transform ($2 \times \mu(t) - 1$) was applied. Finally, we applied a mild gaussian-window smoothing and down-sampling by a factor of ten to discard high-frequency noise that the model not intend to explain while improving the computational efficiency.

Conversely, to project state-space dynamics back into observation (EEG) space, we reconstruct a low-rank spectrogram using the linear interpolation ($\mu(t)U_w^1 + (1 - \mu(t))U_s^1$) (Fig 2a-bottom).

## Model structure

To model the low-dimensional dynamics $\mu(t)$, we propose the following minimally-parameterized first-order model:

$$dx_t = -\left(x_t + 1\right)\left(x_t - \beta(t)\right)\left(x_t - 1\right)dt + \sigma\,dW_t,$$
$$\beta(t) = \tanh\left(\alpha\left(t - t_0\right)\right). \tag{1}$$

The system has by construction two stable fixed points at $x = \pm 1$ and a single unstable fixed point, whose location is determined by the time-varying parameter $\beta(t)$. Hence, the temporal variation in $\beta(t)$ provides a simple mechanism for smoothly shifting the basin boundary between the two attractors. We interpret the $\beta(t)$ as a local proxy for the net sleep drive in biophysically grounded wake-sleep models [16], where slowly varying circadian and homeostatic components act as continuous modulators of an underlying switch. Because we focus on relatively short SOP windows, a smooth hyperbolic tangent function is natural to capture the qualitative features in practice (an approximately stable pre-onset epoch, a finite transition window, and post-onset saturation). The $\beta(t) = tanh(\alpha(t - t_0)$ is the simplest smooth, bounded parameterization of this behavior, with two interpretable parameters, $t_0$ (setting the onset of transition) and $\alpha$ (controlling the changing rate of the sleep-drive). We did not adopt a rectified-linear ramp because it is non-saturating unless explicitly capped, and we did not introduce an asymmetric form because additional flexibility is not required for our aims and would complicate interpretation.

## Parameter inference

The extracted embedding $\mu_{0:N}$ is assumed to be the latent state trajectory generated from stochastic dynamical system (SDE) corrupted by additive gaussian observation noise:

$$\mu_i = x_i + \eta_i, \qquad \eta_i \sim \mathcal{N}(0, \sigma_{\text{obs}}^2). \tag{2}$$

The latent state $x_{0:N}$ itself evolves according to the cubic SDE (Eq. 1) with drift parameters $\theta_{drift} = (\alpha, t_0)$ and diffusion parameter $\theta_{diff} = \sigma$. The EEG recordings and the simulator don't necessarily share the same physical clock: the effective sampling may be slower or faster than the nominal integration steps used in the SDE step. We therefore introduce a time-scale factor, $t^* = \varepsilon t$, which stretches ($\varepsilon > 1$) or compresses (($\varepsilon < 1$) the model time axis so that the latent state can align with an unknown true time-scale of experimental data. Hence, the complete model unknowns are:

$$\theta = \left(\theta_{\text{drift}}, \theta_{\text{diff}}, \varepsilon, \sigma_{\text{obs}}\right) = (\alpha, t_0, \sigma, \varepsilon, \sigma_{\text{obs}}). \tag{3}$$

According to Bayes' rule, the joint posterior density function over parameters and latent path is proportional to

$$\widetilde{\mathcal{L}}(\theta, \mathbf{x}) = \log \pi(\theta) + \sum_{i=0}^{N} \log \mathcal{N}(y_i \mid x_i, \sigma_{\text{obs}}^2)$$

$$+ \sum_{i=0}^{N-1} \log \mathcal{N}\left(x_{i+1} \;\middle|\; x_i + \varepsilon\, f(x_i; \theta_{drift})\, \Delta t, \; (\theta_{diff}\sqrt{\varepsilon})^2\, \Delta t\right), \tag{4}$$

where $\pi(\theta)$ is the joint prior, and $f(x_i; \theta_{drift}) = -(x_i - 1)(x - \beta_i)(x_i + 1)$, $\beta_i = tanh(\alpha\varepsilon(t_i - t_0))$

The three parameters we are interested in are $\{\alpha, t_0, \sigma\}$. For $\alpha$, its prior is set to a half-normal distribution with unit standard deviation ($\alpha_{prior} \sim \mathcal{N}^+(0, 1)$). This forces the fitting procedure to emphasize small alpha regimes, for the two following reasons: first, the timescale of hyperbolic tangent function is inversely proportional to $\alpha$, which means its changing rate is mostly sensitive to the change of $\alpha$ when it's small, and unidentifiable for large $\alpha$ (Fig 7-d). Second, it's a reasonable assumption that the slow drive (a combination of homeostatic pressure and circadian rhythm) evolves in a relatively slow scale. The prior of $t_0$ is set as normal distribution, $t_{0_{prior}} \sim \mathcal{N}(50, 10)$ (simulated case) or $t_{0_{prior}} \sim \mathcal{N}(\frac{N}{2}, \frac{N}{4})$ (experimental data). Only three $t_0$ values were tested in simulated case, because, first, defining highly standardized initial time across subjects is difficult in practical real-data settings and, second, increasing the number of distinct $t_0$ values further increases a lot of computational cost by requiring more loops. The prior distribution of $\sigma$ was $\sigma_{prior} \sim \mathcal{N}^+(0, 2)$ in both simulated and real data to force MCMC only sample the positive values with a relatively broad range. The prior of the time-scale factor, $\varepsilon$, is set as a uniform distribution, $\varepsilon_{prior} \sim \mathcal{U}(0.1, 5)$, which allows the fitted dynamics to be up to ten times slower ($\varepsilon = 0.1$) or five times faster ($\varepsilon = 5$) than the EEG step size. The lower bound prevents numerical stiffness, while the upper bound still covers all physiologically plausible transition speeds encountered across individuals. The inferred posterior distributions of $\varepsilon$ can be found in **Fig H in S1 Appendix**. For observational noise, we assumed a Gaussian observation noise $\mathcal{N}(\mu, \sigma_{obs})$, where $\mu$ is given by the latent state simulated from SDE. In the simulated case, $\sigma_{obs}$ is set as 0.00001 to force the latent state trajectory simulated in MCMC to match the observed simulated trajectory almost exactly. In the real data case, $\sigma_{obs} \sim \mathcal{U}(0.1, 0.4)$ to allow the latent states to differ from the observed data and adapt to potentially different noise-levels to different individuals. The detailed parameter settings can be found in Tables 1, 2 (simulated data) and Table 3 (experimental data).

## Model evaluation

**Simulated data.** To evaluate the parameter inference on simulated data, we simulated 30 random trajectory for every value of $\alpha \in [0.05, 3]$ (logarithmically-spaced), $t_0 \in \{40, 50, 60\}$ and and $\sigma \in \{0.2, 0.5, 1.0\}$, and reported statistics (mean, 95% confidence intervals on the man) of the parameters recovered with MCMC estimation.

**Table 1. Fixed–landscape scenario: Smulation settings and priors.**

| Specification | Value(s) | Prior |
|---|---|---|
| Number of trajectories | 30 | —— |
| Initial state $X_0$ | 0 | —— |
| Unstable fixed-point location $\beta$ | [−0.8:0.2:0.8] | $\mathcal{N}(0, 1)$ |
| Noise $\sigma$ | [0.2:0.1:1.0] | $\mathcal{N}^+(0, 2)$ |

**Table 2. Time-varying landscape scenario: Simulation settings and priors.**

| Specification | Value(s) | Prior |
|---|---|---|
| Number of trajectories | 30 | —— |
| Initial state $X_0$ | 1 | —— |
| Slope $\alpha$ | 10 log-spaced values in [0.05,1], plus 1.5 | $\mathcal{N}^+(0, 1)$ |
| Onset $t_0$ | {40,50,60} | $\mathcal{N}(50, 10)$ |
| Noise $\sigma$ | $\sigma \in \{0.2, 0.3, \ldots, 1.0\}$ | $\mathcal{N}^+(0, 2)$ |

**Table 3. Priors adopted for real EEG data.**

| Parameters | Prior |
|---|---|
| $\alpha$ | $\mathcal{N}^+(0, 1)$ |
| $t_0$ | $\mathcal{N}(\frac{N}{2}, \frac{N}{4})$ |
| $\sigma$ | $\mathcal{N}^+(0, 2)$ |
| $\varepsilon$ | $\mathcal{U}(0.1, 5)$ |
| $\sigma_{obs}$ | $\mathcal{U}(0.1, 0.4)$ |

**Experimental data: RMSE and KL.** To evaluate the parameter inference for real EEG data, we compared the original data with simulations of the model with inferred parameters using both Root Mean Square Error (RMSE) and Kullback-Leibler (KL) divergence (the latter, computed on distributions pooled across all time points). In Fig 8, we applied posterior predictive check to compare between what the fitted model predicts and the actual observed data via running forward simulation of 4000 trajectories from the joint posterior distribution and selected the one with minimal Root Mean Square Error (RMSE) value as well as the one maximizing joint posterior density, then plot its time course and corresponding reconstructed time-frequency plot to compare with the SVD embedding, taken as ground truth. Example posterior draws with minimal RMSEs and distributions of RMSE and KL divergence for all individuals in the test dataset can be found in **Figs I-J in S1 Appendix**.

**Per-timepoint likelihood.** Beyond these two metrics, we also compute the per-time likelihood as this can be informative of how the MCMC procedure manages the compromise between true model noise ($\sigma$) and observation noise ($\sigma_{obs}$), and more generally, goodness of fit. Concretely, Eq. 5 defines the one-step predictive density of the next observation $\mu_{t+1}$ given the current sampled latent state $x_t$ and parameters $\theta$ obtained by marginalizing over the unobserved next state $x_{t+1}$ across posterior draws.

$$p(\mu_{t+1} \mid x_t, \theta) = \int \underbrace{p(\mu_{t+1} \mid x_{t+1}, \sigma_{obs})}_{\text{observation model}} \underbrace{p(x_{t+1} \mid x_t, \sigma, \alpha, t_0, \epsilon)}_{\text{transition model}} \, dx_{t+1}.$$

(5)

For a given draw $s$ (parameters $\theta^{(s)}$, latent state $(x_t^{(s)}, x_{t+1}^{(s)})$), the complete one-step joint density can be decomposed into one observation term and one transition term by taking the logarithm:

$$\log p\left(\mu_{t+1}, x_{t+1} \mid x_t, \theta^{(s)}\right) = \log p\left(\mu_{t+1} \mid x_{t+1}, \sigma_{obs}^{(s)}\right)$$
$$+ \log p\left(x_{t+1} \mid x_t, \alpha^{(s)}, t_0^{(s)}, \sigma^{(s)}, \epsilon^{(s)}\right).$$

(6)

The observation model in Eq. 6 is a Gaussian distribution with mean equal to the latent state and variance $(\sigma_{obs}^{(s)})^2$

$$p\left(\mu_{t+1} \mid x_{t+1}, \sigma_{obs}^{(s)}\right) = \frac{1}{\sqrt{2\pi}\, \sigma_{obs}^{(s)}} \exp\left(-\frac{\left(\mu_{t+1} - x_{t+1}\right)^2}{2\left(\sigma_{obs}^{(s)}\right)^2}\right).$$

(7)

The transition density in Eq. 6 comes from Euler-Maruyama discretization of the cubic SDE. For one given draw $s$, the one-step transition is conditionally Gaussian with mean given by the drift updates and variance set by the diffusion term:

$$m_t^{(s)} := x_t + \epsilon^{(s)} f\left(x_t, t; \alpha^{(s)}, t_0^{(s)}\right) \Delta t,$$
$$v_t^{(s)} := (\sigma^{(s)})^2\, \epsilon^{(s)} \Delta t,$$

(8)

$$p\left(x_{t+1} \mid x_t, \alpha^{(s)}, t_0^{(s)}, \sigma^{(s)}, \epsilon^{(s)}\right) = \frac{1}{\sqrt{2\pi v_t^{(s)}}} \exp\left(-\frac{(x_{t+1} - m_t^{(s)})^2}{2v_t^{(s)}}\right).$$

(9)

These two likelihood terms typically trade off: if $\sigma_{obs}$ is set at a larger value, the fitting procedure allows the state $x_t$ to deviate more from the observation $\mu_t$, and to better fit cubic dynamics; setting $\sigma_{obs}$ at a smaller value constrains the state to fit the observation, at the possible expense of the transition likelihood. See **Text A in S1 Appendix** for an analysis of the detailed timecourse of likelihood within participant.

**Number of switches.** Finally, to analyse the model's behaviour in terms of bistable jumps (Fig 5), as well as evaluate whether parameter inference preserves that behaviour (Fig 10), we computed the number of switches in both simulated trajectories and the real embeddings ($\mu(t)$) with a robust zero-crossing detection rule. First, we smoothed the input (moving-average filter) to suppress high-frequency jitter that would create spurious crossings. We then assign a binary state by the sign of the smoothed signal with a small hysteresis interval around zero (so values within [-0.1,0.1] keep the previous state rather than triggering a switch). A switch is counted only if the trajectory commits to the new sign for at least 50 consecutive samples.

## Supporting information

**S1 Appendix. Text A. Analysis of the timecourse of observation and cubic transition likelihoods within individuals. Text B.** Affine link between $\mu(t)$ and $V_2$. **Fig A.** Explained variance of SVD modes in wake and sleep segments. **Fig B.** The reported association between cubic parameters and subjective sleepiness is not driven by participants who have low model fit. **Fig C.** Parameter–behavior associations remain consistent across $\sigma_{obs}$ settings. **Fig D.** Comparison of embeddings with Li et al. (2025). **Fig E.** First singular modes ($U^1$) of a global SVD analysis of the full SOP spectrogram, for all participants in our test dataset. **Fig F.** Temporal weights of the first singular modes ($V_{(:,1)}^T$), superimposed to total spectral power. **Fig G.** Spectrogram localization of SOP windows for all participants. **Fig H.** Posterior distributions of $\epsilon$. **Fig I.** Example posterior draws with minimal RMSE. **Fig J.** Distribution of KL-divergence and RMSE over posterior samples. (PDF)

## Acknowledgments

We thank Björn Rasch, Thomas Andrillon, Megan Morrison, and Lou Zonca for useful discussions on previous versions of this work.

## Author contributions

**Conceptualization:** Zhenxing Hu, J. Nathan Kutz.

**Data curation:** Zhenxing Hu.

**Formal analysis:** Zhenxing Hu.

**Funding acquisition:** J. Nathan Kutz, Jean-Julien Aucouturier.

**Methodology:** Zhenxing Hu, Manaoj Aravind, J. Nathan Kutz, Jean-Julien Aucouturier.

**Project administration:** Jean-Julien Aucouturier.

**Resources:** Xu Lei, Jean-Julien Aucouturier.

**Software:** Zhenxing Hu.

**Supervision:** J. Nathan Kutz, Jean-Julien Aucouturier.

**Validation:** Zhenxing Hu, Jean-Julien Aucouturier.

**Visualization:** Zhenxing Hu.

**Writing – original draft:** Zhenxing Hu.

**Writing – review & editing:** Manaoj Aravind, J. Nathan Kutz, Jean-Julien Aucouturier.

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
