## [Decision Letter · Decision Letter 0]

3 Dec 2025

PCOMPBIOL-D-25-01784

Learning the bistable cortical dynamics of the sleep-onset period

PLOS Computational Biology

Dear Dr. Hu,

Thank you for submitting your manuscript to PLOS Computational Biology. After careful consideration, we feel that it has merit but does not fully meet PLOS Computational Biology's publication criteria as it currently stands. Therefore, we invite you to submit a revised version of the manuscript that addresses the points raised during the review process.

We look forward to receiving your revised manuscript.

Kind regards,

Paul Bays

Academic Editor

PLOS Computational Biology

Andrea E. Martin

Section Editor

PLOS Computational Biology

**Additional Editor Comments:**

As you will see, the Reviewers have positive things to say about your manuscript but also raised some concerns that will need to be addressed before a final decision can be made.

**Journal Requirements:**

3) Please amend your detailed Financial Disclosure statement. This is published with the article. It must therefore be completed in full sentences and contain the exact wording you wish to be published.

State what role the funders took in the study. If the funders had no role in your study, please state: "The funders had no role in study design, data collection and analysis, decision to publish, or preparation of the manuscript.".

**Reviewers' comments:**

Reviewer's Responses to Questions

**Comments to the Authors:**

Reviewer #1: In this work, Chen and colleagues proposes a dynamical systems model of sleep onset and a corresponding parameter inference procedure to extract per-individual sleep parameters from EEG data. The model is based on stochastic double-well dynamics with a monotonic sleep drive that shifts the potential function. The model parameters are fit to a 1D embedding of EEG spectral mode differences between wake and sleep. The authors validate the inference procedure on simulations, and apply it on real EEG data. The inference procedure is generally accurate, produces faithful reconstructions of the spectrogram, and results in parameter estimates that are correlated with subjective reports of sleepiness.

In general, the model seems to me well-motivated, simple, and elegant. The double-well dynamics is a natural choice for such a two-state transition between wake and sleep, and inferring the model parameters based on a low-dimensional decomposition of EEG spectrogram appears reasonable as well. The MCMC-based inference procedure works reasonably well, though where it’s less accurate may be important to consider (see comment below). The application on real EEG recordings and the resulting association with sleepiness are interesting and suggest future potential in linking dynamics at the phenomenological level to underlying physiology though the authors are careful in not speculating specific neural substrates. Overall, the paper was clearly structured, written, and nicely illustrated, especially the schematic of the workflow in Figure 1.

I don’t have major concerns in the technical components of the work, but mainly about the assumptions and interpretations, and some clarification questions about methodological considerations:

- the 2-state dynamic makes sense considering wake to sleep, but what about the other sleep stages? They have distinct spectral signatures, while some states (like REM) could be considered to be in between wake and sleep. Related, one would intuitively think that such a model, in reverse, could be used to explain sleep to wake transitions, though I don’t think such a symmetry is present?

- another point worth discussing are sensory disturbances in the process of falling asleep, including e.g., self-motion.

- the spectrogram representation of the EEG ignores phase, in particular coupling across frequencies, which could be an informative marker (e.g., up-down states). Are any such considerations relevant in the context of this model? Or is the goal simply to look for maximum SNR to separate the two stages?

- what motivates the form of the transition as a tanh, as opposed to, e.g., rectified linear, or an asymmetric function?

- Related, is there a way to retrieve / infer the wake and sleep modes in an unsupervised fashion, e.g., without manually labeling the wake and sleep periods? Another way to conceptualize it is to fit the spectral mode (difference), along with the transition and noise parameters, directly and jointly from the spectrogram data? The discussion mentions the second SVD component of the overall spectrogram, but in the context of a forced energy transfer between the top sleep and wake modes.

- time in the non-stationary simulations (e.g., Fig 6 a and d) is on the order of ~50seconds, but in the real data the transition appears to happen in about 10x longer timescale (400-500s), and with variable absolute duration (N). Are the conclusions from the simulated scenarios still valid given this discrepancy, especially since the inference procedure tends to overestimate alpha at lower values (corresponding to longer timescale transition), or is it simply a matter of rescaling? It would be informative to see what the non-stationary simulations in the “realistic” timescale look like, as well as the fitted posterior predictive samples, as in Fig 7c and d.

- is KL computed on distributions collapsed over time (e.g., Fig 7b)? In addition to KL and RMSE, because the trajectories have an additive stochastic component in sigma assumed to be a Gaussian, one could in theory compute the per-timepoint data likelihood under the fitted models.

- what do some of the worse / non-minimal RMSE/KL samples look like?

- in Figure 8, estimated alpha (transition slope) and sigma (noise level) are both found to correlate with pre-nap sleepiness, with potential proposed interpretations for both. It’s intuitive that the speed of transition into sleep should correlate with sleepiness, but the correlation with noise amplitude is interesting and surprising. One potential explanation that’s not necessarily “neural” is that the inference of these two parameters jointly results in a correlation between the two, i.e., there is degeneracy between the two parameters given an observation. Is this typically observed, e.g., in inferred parameters from simulated data? Concretely, looking at the joint distribution of inferred values in Fig 6 for example, are there systematic correlations between the values (and errors) of the different parameters?

- I’m not sure I understood what is meant by “how possibly” / phenomenon models (line 44-46). Based on the explanation that follows, that means non-mechanistic / non-neural, or specifically that their parameters were not estimated per-individual? I’m asking this in the context of clarifying the contribution here, and whether the advancement is in a new model, a parameter estimation procedure, or both.

- related, one stated contribution of the model and associated inference workflow is that the inferred model parameters may shed light on the mechanisms of sleep onset, as shown by its correlation with sleepiness report. In other words, they may serve as model-based biomarkers for sleep stages and disorders. While this is true, are there simpler metrics one can compute from the data itself, without the complex MCMC inference procedure, that correlates well (or better) with the subjective ratings? Alternatively, could one potentially make physiological interpretations of the parameters, which would provide additional mechanistic and treatment value beyond data-driven metrics (though it is explicitly discussed in the limitations that b(t) is not necessarily physiologically grounded). If not, how should one view the current model beyond a “reanalysis” of the EEG data? In general, while the model is intuitive and elegant, I think the stated contribution section (lines 44-54) is a bit narrow in scope.

- there are some previous works on decomposing EEG spectrograms specifically via SVD / PCA from KJ (Kai) Miller that are probably worth discussing and relating to (around line 67)

Note that: 1) I’m not well-versed in sleep so I cannot judge the novelty and significance of the contribution with respect to that literature, 2) with respect to COI, we have had some previous discussion about this work (and apologies if I raised the same annoying points as before and simply forgot).

Richard Gao, PhD

Goethe University Frankfurt

Reviewer #2: The manuscript presents a significant and novel contribution to the computational modeling of sleep regulation. It moves beyond the traditional view of sleep as an abrupt step-function, proposing a minimally-parameterized stochastic dynamical model that accurately captures the non-monotonic, “flickering” dynamics characteristic of the sleep-onset period (SOP). By modeling the transition as a noisy, slowly tilting bistable potential landscape, the authors provide an elegant framework to estimate two crucial, person-specific parameters: the rate of landscape change (alpha) and the intrinsic noise level (sigma). Most importantly, the study validates the relevance of this mechanistic approach by showing that these estimated dynamic parameters correlate significantly with subjective measures of sleepiness, thereby offering novel, quantitative biomarkers for characterizing individual variability in the complex process of falling asleep.

While this study offers a promising theoretical framework of sleep-onset transition, several major and minor issues related to model validation, novelty in light of recent literature, and data interpretation must be fully addressed before the manuscript can be reconsidered for publication. My specific comments are detailed below.

Major concerns

1. A paper published last month (Li, Junheng, Anastasia Ilina, Robert Peach, et al. “Falling Asleep Follows a Predictable Bifurcation Dynamic.” Nature Neuroscience, October 28, 2025, 1–11. https://doi.org/10.1038/s41593-025-02091-1) also uses a fold bifurcation model to describe the sleep-onset process. The readers will inevitably compare this paper with the authors’ work. The authors must introduce a section, likely in the Discussion, that compares their model to other approaches and explains why their model is superior for their task.

2. The authors mentioned the reduced Philipps-Robinson model (reference #15), which is probably the first model that uses bifurcation to quantify sleep-wake transition. Is the authors’ model mathematically equivalent to the reduced PR model (in terms of the bifurcation structure)? If it does, can authors identify the biological components in the reduced (or the original) PR model that correspond to the parameters (alpha, beta, sigma, etc) in the proposed model and explore what biological processes contribute to “flickering?” The manuscript is submitted to PLoS Computational “biology,” hence a bit more discussion in biology is highly appreciated.

3. The manuscript correctly notes in the Discussion that the 1D SVD-based embedding leverages the core idea of mutual inhibition and that its limitation is missing nuanced amplitude fluctuations. However, the justification for the sufficiency of this reduction relies on a statement that “this 1-D scheme suffices to capture fundamental wake-sleep transitions” without providing the empirical support to back this statement. To rigorously validate the model’s foundation, the author must quantify the explained variance of the first SVD mode across the entire cohort. More importantly, the authors should provide evidence that the second (or subsequent)SVD modes do not capture dynamics critical for the wake-sleep transition or that incorporating them does not significantly improve the fit or correlation with the subjective sleepiness scores.

4. Figure 6c basically shows that the parameter alpha cannot be accurately estimated from individual data. But in Figure 8 left, the authors tried to show that the estimated alpha is correlated with an individual’s report of sleepiness. If the alpha values are not accurate, how reliable is the figure 8? I can see (from Fig. 6c) that there are monotonic errors between the estimated and true alpha values. So perhaps the estimated alpha can be corrected by a simple math equation within a certain range? If so, the authors can use the “corrected” alpha values in figure 8.

5. Following the last comment, is MCMC the best method to estimate parameters? It is even better if the authors can find a better method to estimate alpha without the post-estimate correction suggested above.

6. Figure 8 right. Why is the noise level (sigma) correlated with reported sleepiness? In the manuscript, the authors explained that a higher sigma can lead to earlier sleep onset. However, a higher sigma may also produce more “sleep-to-wake” transitions, which should not happen for a person with a high sleepiness level. The authors should analyze their simulated and experimental data and examine this issue. Maybe the authors can also explain it from a biological perspective. What biological processes contribute to sigma (see my major comment #2) and whether they make sense.

Minor concerns

1. Line 35. “saddle-point bifurcation” should be “saddle-node bifurcation.”

2. Figure 2a. I do not see “dashed lines” mentioned in the main text and figure legends.

3. Model equations are repetitively shown in Results and Methods. The authors should just choose one place (preferably in the Methods) to show and explain.

4. Line 109 (eq 1). Replace x’ with dx/dt for clarity.

5. Figure 5 right. Why did the authors only test three sigma values? Can they test more sigma values, similar to the alpha values in Figure 5 left, to make the trends of the curve clearer?

6. Figure 4 right and Figure 5b & 5c. The plots used to compare estimated and true parameter values (specifically for alpha in Fig. 5C) are confusing and misleading due to the use of a nonlinear x-axis scale. The rigorously demonstrate the accuracy of the parameter estimation procedure, the author must replot these data using linear x and y aces with the same range. This should include a reference 45 degree line (y=x) to allow readers to visually and quantitatively assess the magnitude of the estimation error (i.e. how “off” the estimated values are from the true values).

7. Line 396. Were all EEG channels used when the authors applied SVD analysis? More details should be given about the SVD analysis.

**Have the authors made all data and (if applicable) computational code underlying the findings in their manuscript fully available?**

Reviewer #1: Yes

Reviewer #2: Yes

PLOS authors have the option to publish the peer review history of their article (what does this mean?). If published, this will include your full peer review and any attached files.

Reviewer #1: **Yes:** Richard Gao

Reviewer #2: **Yes:** Chung-Chuan Lo

**Figure resubmission:**
---

## [Decision Letter · Decision Letter 1]

27 Mar 2026

PCOMPBIOL-D-25-01784R1

Learning the bistable cortical dynamics of the sleep-onset period

PLOS Computational Biology

Dear Dr. Hu,

Thank you for submitting your manuscript to PLOS Computational Biology. After careful consideration, we feel that it has merit but does not fully meet PLOS Computational Biology's publication criteria as it currently stands. Therefore, we invite you to submit a revised version of the manuscript that addresses the points raised during the review process.

We look forward to receiving your revised manuscript.

Kind regards,

Paul Bays

Academic Editor

PLOS Computational Biology

Andrea E. Martin

Section Editor

PLOS Computational Biology

**Additional Editor Comments :**

Please address the minor comments from Reviewer 2.

**Reviewers' comments:**

Reviewer's Responses to Questions

Reviewer #1: Many thanks to the authors for constructively engaging with my comments and a comprehensive revision. I especially appreciated the new results with SINDY, the factored likelihood model, and the parameter degeneracy analysis, as well as the clarified contribution in general and relative to the new Li et al. 2025 paper.

The authors have addressed all my concerns adequately and I think this is a fine and interesting piece of work on the role of potential neural noise mechanisms for sleep onset.

Richard Gao, PhD

Goethe University Frankfurt

Reviewer #2: I appreciate the authors’ thorough efforts in addressing my previous comments. The additional data and plots significantly strengthen their arguments, and I have no further major concerns regarding the scientific content.

However, upon reviewing the revised figures, I identified several inconsistencies and formatting issues that need to be addressed before publication. I recommend a careful final check of all figures and legends. My specific observations are as follows:

1. Figure 3: The panel labels "a)" and "b)" are missing.

2. Figure 6 (Right): Regarding my previous comment about the "limited sample size" (formerly Fig 5), the authors acknowledged the issue but updated Fig 7B instead. The specific panel I mentioned (now Figure 6 right) remains unchanged with only three data points. Please ensure this is addressed.

3. Figure 7: Panel labels "A–D" are missing from the graphics.

4. Figure 7 caption: There is a typographical error in the caption where "Panel C" is cited twice; one of these references should be changed to Panel “B.”

5. Figure 7, Panel B: There is a discrepancy between the plot and the caption. The plot displays data for nine sigma values, whereas the caption only specifies three.

6. Consistency in Labeling: There is an inconsistency in the panel labeling style throughout the manuscript. Some figures use lowercase letters (e.g., a, b) while others use uppercase (e.g., A, B). Please ensure a consistent labeling format is applied to all figures.

**Have the authors made all data and (if applicable) computational code underlying the findings in their manuscript fully available?**

Reviewer #1: Yes

Reviewer #2: Yes

PLOS authors have the option to publish the peer review history of their article (what does this mean?). If published, this will include your full peer review and any attached files.

Reviewer #1: **Yes:** Richard Gao

Reviewer #2: **Yes:** Chung-Chuan Lo

**Figure resubmission:**
---

## [Editor Report · Decision Letter 2]

17 Apr 2026

Dear Hu,

We are pleased to inform you that your manuscript 'Learning the bistable cortical dynamics of the sleep-onset period' has been provisionally accepted for publication in PLOS Computational Biology.

Best regards,

Paul Bays

Academic Editor

PLOS Computational Biology

Andrea E. Martin

Section Editor

PLOS Computational Biology

---

## [Editor Report · Acceptance letter]

PCOMPBIOL-D-25-01784R2

Learning the bistable cortical dynamics of the sleep-onset period

Dear Dr Hu,

I am pleased to inform you that your manuscript has been formally accepted for publication in PLOS Computational Biology. Your manuscript is now with our production department and you will be notified of the publication date in due course.

With kind regards,

Anita Estes
